# Durability of Slippery Liquid-Infused Surfaces: Challenges and Advances

Divyansh Tripathi [1], Prauteeto Ray [1], Ajay Vikram Singh [2], Vimal Kishore [1,*] and Swarn Lata Singh [3,*]

[1] Department of Physics, Banaras Hindu University, Varanasi 221005, India; divyanshtripathi.bsk@gmail.com (D.T.); prauteetoray@gmail.com (P.R.)
[2] Department of Chemical and Product Safety, German Federal Institute of Risk Assessment (BfR), Maxdohrnstrasse 8-10, 10589 Berlin, Germany; ajay-vikram.singh@bfr.bund.de
[3] Department of Physics, Mahila Mahavidyalaya (MMV), Banaras Hindu University, Varanasi 221005, India
[*] Correspondence: vimalk@bhu.ac.in (V.K.); swarn@bhu.ac.in (S.L.S.)

**Abstract:** Slippery liquid-infused porous surfaces (SLIPS) have emerged as a unique approach to creating surfaces that can resist fouling when placed in contact with aqueous media, organic fluids, or biological organisms. These surfaces are composed of essentially two components: a *liquid lubricant* that is locked within the protrusions of a *textured solid* due to capillarity. Drops, immiscible to the lubricant, exhibit high mobility and very-low-contact-angle hysteresis when placed on such surfaces. Moreover, these surfaces are shown to resist adhesion to a wide range of fluids, can withstand high pressure, and are able to self-clean. Due to these remarkable properties, SLIPS are considered a promising candidate for applications such as designing anti-fouling and anti-corrosion surfaces, drag reduction, and fluid manipulation. These collective properties, however, are only available as long as the lubricant remains infused within the surface protrusions. A number of mechanisms can drive the depletion of the lubricant from the interior of the texture, leading to the loss of functionality of SLIPS. Lubricant depletion is one challenge that is hindering the real-world application of these surfaces. This review mainly focuses on the studies conducted in the context of enhancing the lubricant retention abilities of SLIPS. In addition, a concise introduction of wetting transitions on structured as well as liquid-infused surfaces is given. We also discuss, briefly, the mechanisms that are responsible for lubricant depletion.

**Keywords:** wetting; slippery liquid-infused porous surfaces; anti-fouling surfaces; lubricant depletion; durability of SLIPS

## 1. Introduction

Liquid–solid interfaces are ubiquitous: from the interplay between membranes and the cellular bulk to designing anti-fouling surfaces, the wettability of the surface plays an important role. The ability to tune the wettability of a surface is critical in a broad range of industrial, technological, and biomedical applications. The requirement for surface wettability varies from one application to the other. In applications such as painting and printing [1], pharmaceuticals, and the agro-chemical industry, good surface wettability is required [2], whereas liquid-repellent surfaces form a basis for the design of anti-fogging, anti-icing, anti-corrosion, and anti-biofouling surfaces [3]. Unwanted accumulation on surfaces, when in contact with an aqueous medium, leads to problems affecting almost all facets of life. The fogging and icing of surfaces and the failure of materials due to corrosion and contamination are some examples that are well known. Biofouling, a result of the undesired adhesion of microorganisms to surfaces, is another critical issue in a wide range of scenarios. From medical devices [4] and the food industry [5] to the supply of drinking water [6], the biofouling of surfaces poses a serious threat to human health [7]. Marine biofouling is another challenge that affects both the ocean economy and the environment. The accumulation of sea organisms on artificial surfaces immersed in water results in an

increase in weight and drag, which, in turn, affects a range of industries, from oil and gas to marine renewable energy and shipping [8,9].

The corrosion and fouling of surfaces, in various settings, lead to huge economic losses, health hazards, and environmental damages. It is estimated that the economic loss due to corrosion represents up to 3%–5% of the gross domestic product of industrialized countries [10]. Moreover, a substantial amount of packaged product is lost due to adhesion to the walls of the container. It is estimated that up to 15% of packaged food (especially viscous products such as honey) remains stuck to the packaging material surface [11,12]. Ice formation on outdoor surfaces is a similar issue, adversely affecting multiple industries, such as transportation, agriculture, energy, and construction [13–15].

The wettability of a surface is thus an essential factor in designing anti-fouling surfaces. Whether a liquid will wet a given surface or not depends upon the liquid–solid interaction. This interaction can be tuned in two ways. The first and lesser-explored method involves the modification of the properties of the liquid [2,16–19]. Modifying the chemical and physical properties of the surface remains the most exploited method of controlling the wetting behavior at an interface. The modification of the surface properties to achieve liquid repellency is inspired by lotus leaves [20]. Figure 1 shows the surface of a lotus leaf as obtained by Barthlott and Neinhuis using a scanning electron microscope [21] (Figure 1). As can be seen, the leaf consists of a combination of two scales of roughness. The leaf surface has a layer of micron-sized papillae that are covered with wax nanocrystalloids. The wax itself is not superhydrophobic but, when combined with hierarchical micro-nanostructures, the combination renders the lotus leaves superhydrophobic and offers self-cleaning properties. Typically, the fabrication of a superhydrophobic surface involves the construction of a rough surface (micro-/nanometer-sized hierarchical structures) followed by coating with a low-surface-energy material. These coatings, however, come with their own disadvantages, their durability and toxicity being the main concerns. Recently, there has been a great deal of effort to create environmentally friendly superhydrophobic coatings, especially for applications such as biomedical devices and the marine and food industries. While beyond the scope of this review, we mention here a few articles discussing the current status of these coatings [5,22–28].

The other critical issue concerning lotus leaf mimics is their durability. The roughness plays a critical role in minimizing the contact between the solid surface and the external liquid by trapping air pockets within the surface protrusions. However, these air pockets are highly compressible and cannot withstand pressure, leading to a loss of liquid repellency under increased pressure or upon impact. The challenges increase further if the surface comes into contact with a low-surface-tension liquid, such as organic liquids [29–31]. Furthermore, especially for microstructured surfaces, the liquid repellency cannot be easily restored if lost. It was proposed that adding nanoscale roughness could provide liquid-repellent surfaces with greater robustness and durability [32,33]. The risk of liquid sticking to the surface increases further if there are defects, contamination, or damaged sites present on the surface. There are a number of excellent review articles on lotus leaf mimics that give an overview of the challenges, limitations, and important advances made so far; here, we mention a few [34–39].

An alternative route to creating more robust and stable liquid-repellent surfaces utilizes the surface protrusions to lock in a low-surface-tension liquid (lubricant) due to capillarity. If the solid–lubricant surface energies are well matched, and the degree of roughness is chosen properly, the liquid fills in the protrusions and creates a continuous macroscopic film of liquid that covers the solid surface completely. This arrangement renders a smooth, defect-free, and stable interface that is capable of withstanding pressure as the highly compressible air pockets are now replaced with a liquid that is comparatively less compressible [40–42]. Any external liquids that are immiscible with the lubricant can then be easily removed as the external liquid rests on the lubricant layer and not underneath the substrate. This results in very low sliding angles and excellent self-cleaning properties.

SLIPS also exhibit self-healing and are able to maintain repellency for a wide range of liquids [41,43,44].

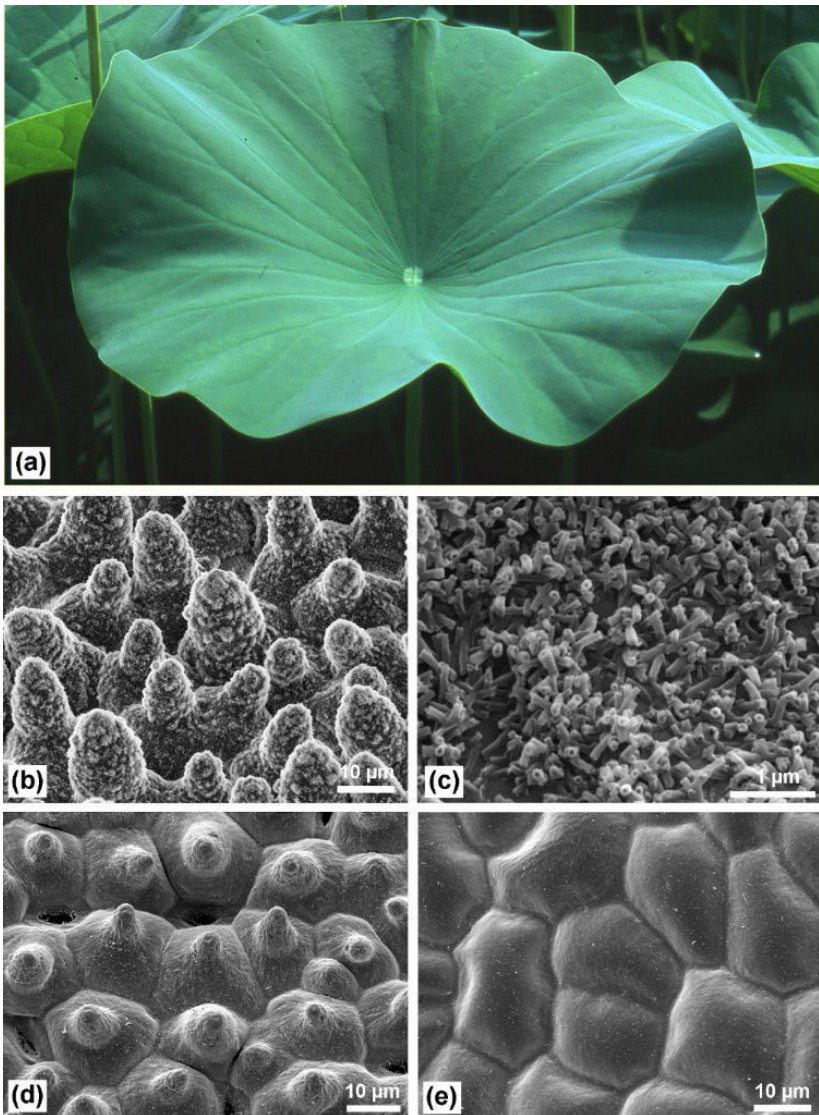

**Figure 1.** (**a**) Lotus leaf, which exhibits excellent water repellency and self-cleaning. (**b**) Scanning electron microscopy (SEM) image of the upper side of the leaf, showing the hierarchical surface structure consisting of micron-sized papillae and nanosized wax crystalloids. (**c**) Nanosized wax rods. (**d**) Upper surface of the leaf after critical point (CP) drying; the wax crystalloids are dissolved and, hence, the stomata are more visible. (**e**) Leaf underside after CP drying, showing the convex cells without stomata. Reproduced from Ref. [21] with permission from Springer Nature.

With these collective properties, liquid-infused surfaces have emerged as a promising candidate for the design of robust surfaces for applications such as anti-icing [44,45], anti corrosion [46,47], anti-biofouling [3,48], and drag reduction [9,44,49]. There are, however, still a few challenges hindering the practical application of liquid-infused surfaces: the greatest limitation is the loss of lubricant due to evaporation, displacement, cloaking, or any other mechanism. A more detailed discussion of lubricant depletion is given in Section 3. Since it is the infused lubricant layer that affords SLIPS their unique and favorable properties, the retention of the lubricant is essential.

In this review article, we will discuss recent studies aimed at improving the durability of liquid-infused surfaces. In the next section (Section 2), we will give a brief account of the slippery liquid-infused porous surfaces. In this section, we will also give a concise review

of the wetting transitions on both the lotus leaf mimics and SLIPS. We will also discuss the design parameters that are required for robust SLIPS. In Section 3, we will discuss, first, the factors responsible for lubricant loss, followed by the recent case studies on the improvement of the durability of liquid-infused surfaces. We will also briefly discuss the mechanisms proposed for lubricant replenishment in order to keep the SLIPS functional. Toward the end of Section 3, we will give a brief introduction of organogels, which are proposed as an alternative to conventional SLIPS. We will conclude the review article by providing a concise summary and discussing opportunities to realize the full potential of liquid-infused surfaces.

## 2. Liquid-Infused Surfaces

A liquid-infused surface consists of an underlying textured solid and a suitable liquid lubricant that can wick into, spread, and adhere stably within the protrusions. The immobilized lubricant creates a slippery interface: any impinging liquid that is immiscible to the lubricant essentially floats on the lubricant, instead of sticking to the substrate, and can be easily removed with a small tilt (see Figure 2). Moreover, as long as the lubricant and the external liquid remain immiscible, the SLIPS maintain their repellency for liquids of even low surface tension. Another unique property possessed by SLIPS is the ability to self-heal, as the lubricant can flow and fill any damaged site on the surface [41,43].

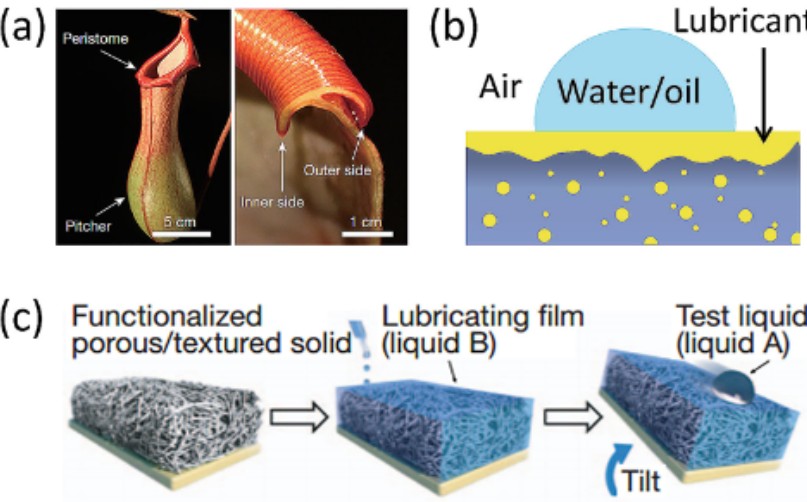

**Figure 2.** Slippery liquid-infused porous surfaces inspired by Nepenthes pitcher plant, as shown in (**a**). The hydrophilic microstructured peristome is utilized to lock in water and create a slippery surface. (**b**) Shows a general scheme of the creation of SLIPS. (**c**) is a schematic representation of the fabrication of a liquid-infused slippery surface. Reproduced from Ref. [48] with permission from John Wiley and Sons.

The finest example of a liquid-infused slippery surface is found in the Nepenthes pitcher plant (see Figure 2); the peristome of the plant is microtextured and hydrophilic. On a wet day, these microstructures are filled with water and a continuous wetting film of the liquid covers the surface. When an insect steps onto the peristome, it slips and falls into the digestive juices [41,50]. The first liquid-infused surfaces were reported in the year 2011, by the groups of J. Aizenberg [41], K. Varanasi [42], and D. Quéré [51]. The idea of such a surface, however, was first introduced briefly by D. Quere in 2005 as a slippery composite surface [40].

Since a liquid-infused surface can function successfully only as long as the lubricant remains infused within the surface texture, the choice of the solid and lubricant should ensure that the lubricant spreads and forms a wetting film. Before continuing with this discussion, we will take a small detour and review the fundamentals of wetting on smooth and textured surfaces.

### 2.1. Wetting of a Solid Surface

When a liquid comes into contact with a solid surface, there are two possibilities: either the liquid spreads and wets the surface or it tries to remain in a spherical shape, minimizing its contact with the lower substrate. The shape that the liquid assumes depends on the intermolecular adhesive interactions between the two phases. Young, in his pioneering work, quantified the degree of contact at a given solid–liquid interface in terms of the static contact angle [52]. Young's (static) contact angle ($\theta_Y$) is found via the in-plane balance of the liquid–solid $\gamma_{SL}$, liquid–vapor ($\gamma_{LV}$), and solid–vapor ($\gamma_{SV}$) interfacial energies at the three phase contact lines (see Figure 3a). The complete wetting regime is characterized by $\theta_Y = 0$. As the contact angle increases, the wettability of the solid surface decreases; it is a common notion that a transition from lyophilic to lyophobic happens at $\theta_Y = 90°$, (or when $\gamma_{SV} = \gamma_{SL}$), and the surface becomes ultraphobic for a contact angle $> 150°$ [53,54]. Young's model assumes a smooth and chemically homogeneous surface; however, the real solids are rough and chemically heterogeneous. The presence of roughness modifies Young's contact angle, as was explained by Wenzel and Cassie. The Wenzel model, which assumes that the liquid closely follows the surface asperity (see Figure 3b), predicts the apparent contact angle $\theta_A$, on a rough surface, as

$$cos\theta_A = rcos\theta_Y \tag{1}$$

where $r$ (a number always larger than unity) is the surface roughness factor, defined as the ratio of the actual over the apparent surface area of the substrate. Since roughness increases the area that the liquid is in contact with, the Wenzel model indicates an enhancement in the material's existing tendency, due to the presence of roughness; a hydrophilic material will become more hydrophilic, whereas a hydrophobic material will become even more hydrophobic. However, for hydrophobic surfaces, the liquid does not follow the surface topography, but remains suspended in the air pockets entrapped within the surface textures, as shown in the scheme in Figure 3c. The contact angle for this mode was given by Cassie [55], which, for the specific case of solid–air composite surfaces, is as follows:

$$cos\theta_A = \phi(cos\theta_Y + 1) - 1$$

where $\phi$ is the area fraction of the solid in contact with the liquid, $1 - \phi$ is the area fraction that is in contact with the air, and $\theta_Y$ is Young's contact angle on a corresponding smooth solid surface. Thus, in Cassie's model, the liquid drop is highly mobile and can roll off easily. A transition from the Cassie to the Wenzel state happens when the apparent contact angle is decreased to a critical value $\theta_c$. $\theta_c$ is obtained by equating the interfacial energies in both the Wenzel and the Cassie states and is given as follows [56]:

$$cos\theta_c = \frac{\phi - 1}{r - \phi} \tag{2}$$

Moreover, there is a possibility of a mixed state where some part of the surface is in the Wenzel state and the remaining part is in the Cassie state [57]. Another useful quantity to characterize a surface is the contact angle hysteresis, which becomes very important when a liquid is in motion on a textured surface, since Young's contact angle is defined in equilibrium. Contact angle hysteresis (CAH) is defined as the difference between the advancing and the receding contact angles (see Figure 3d). The dynamic advancing contact angle ($\theta_{adv}$) is the angle at the three phase contact point upon increasing the liquid volume, whereas the dynamic receding contact angle ($\theta_{rec}$) is the contact angle upon decreasing the liquid volume. Thus, essentially, $\theta_{adv}$ represents the wettability of the surface and $\theta_{rec}$ represents the adhesion on the surface. The CAH quantifies the adhesion and self-cleaning properties of a surface [58]; the lower the CAH, the better the liquid slides on the surface [59]. These models, however, are for surfaces with single-scale textures, whereas

it is found that the hierarchical structures (structures with two or more than two length scales) not only increase the liquid repellency but also stabilize the Cassie state [60–62].

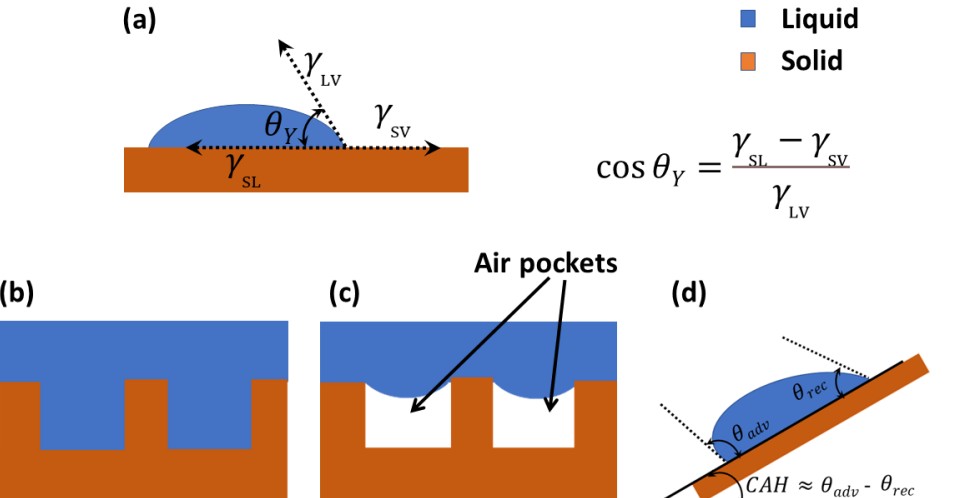

**Figure 3.** (**a**) shows wetting on a smooth surface; $\theta_Y$ is Young's contact angle; $\gamma_{SL}$, $\gamma_{LV}$ and $\gamma_{SV}$ are the interfacial energies of solid–liquid, liquid–vapor and solid–vapor interfaces, respectively. (**b**) shows a liquid in the Wenzel state on a textured surface. (**c**) shows a liquid in the Cassie state. (**d**) shows contact angle hysteresis for a moving drop; $\theta_{adv}$ is the dynamic advancing contact angle and $\theta_{rec}$ is the dynamic receding contact angle.

### 2.2. Wetting of Liquid-Infused Surfaces

To fabricate a liquid-infused porous surface, the lubricant should be chosen such that it infiltrates into, spreads, and adheres stably within the surface corrugations [41]. This can be achieved by appropriately choosing the surface roughness and the lubricant type. At the limit of the vanishing contact angle, the lubricant will form a macroscopic wetting layer and cover the solid surface completely. When this liquid-infused surface comes into contact with an external fluid, depending upon the properties of the external fluid, the lubricant and the solid surface, different morphologies can appear. The equilibrium state is represented by a balance of the interfacial tensions (there are six different interfacial tensions present in a system composed of a SLIP and an external liquid) or by the value of the spreading parameter $S_{ab}(c)$, which is defined as follows [63]:

$$S_{ab}(c) = \gamma_{ab} - (\gamma_{bc} + \gamma_{ac}) \tag{3}$$

where $\gamma_{ab}$, $\gamma_{bc}$ and $\gamma_{ac}$ represent the interfacial surface energies between the $a - b$, $b - c$ and $a - c$ phases, respectively. $S_{ab}(c)$ predicts whether phase $a$ will wet $b$ in the presence of phase $c$; $S_{ab}(c) > 0$ ensures that $a$ will wet $b$ stably in the presence of $c$. For a choice of lubricant and external fluid such that the spreading parameter of the lubricant on the working fluid in the presence of air $S_{ow}(a) > 0$, where $o$, $w$ and $a$ represent the lubricant, external liquid and air, respectively, the lubricant will spread over the top of the external fluid and form a cloak [64] (see Figure 4a). The thickness of the cloak layer is determined by a balance between the Laplace pressure and the repulsive disjoining pressure. Cloaking is one of the factors responsible for the depletion of the lubricant; the shedding of cloaked drops leads to the loss. Moreover, droplets smaller than the protrusions can be submerged and replace the lubricant [65,66]. For cases when $S_{ow}(a) < 0$, the lubricant forms a wetting ridge around the droplet, as can be seen in Figure 4b. We will discuss these wetting ridges in the following.

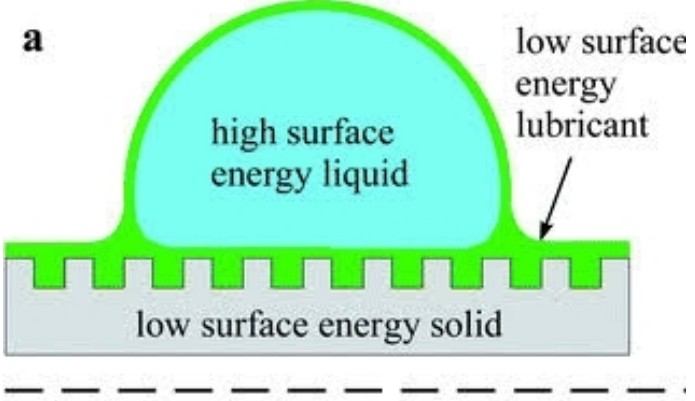

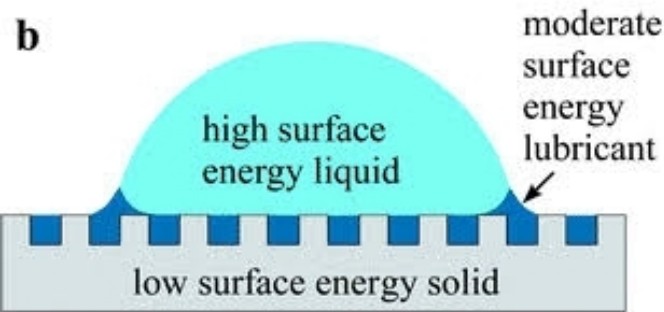

**Figure 4.** (**a**) shows the case of cloaking when $S_{ow}(a) > 0$, and (**b**) corresponds to the case of $S_{ow} < 0$. Wetting ridges can be seen at the contact point. Reproduced from Ref. [63] with permission from the Royal Society of Chemistry.

Depending upon the interfacial energies involved, the lubricant underneath the drop can exist in three different states. In the first state, the working fluid wets the protrusions completely by replacing the lubricant from the interiors of the pores. This state is known as an impaled state. If the lubricant adheres within the protrusions but the external liquid contacts the exposed solid, this is called the impregnated state. In the encapsulated state, the lubricant prevents any contact between the external fluid and the lower solid. These three states are shown as states *W*1, *W*2 and *W*3 in column 2 of Figure 5. Smith et al. [63] used thermodynamic arguments and presented 12 different possible thermodynamic states of a drop on a liquid-infused surface. We show the phase diagram constructed by Smith et al. in Figure 6. The interfacial energies for these states below and outside the drop, in terms of the individual interfacial energy contributions, are given in the Figure 5. Corresponding criteria in terms of the spreading parameters and contact angles are also given in column 4. For more details of the design principles of stable SLIPS, we refer to [63,66–69].

For the cases when $S_{ow}(a) < 0$, the lubricant forms wetting ridges surrounding the working fluid due to capillarity (see Figure 4b). The height of this ridge is determined by a balance between the Laplace pressure and the hydrostatic pressure due to pulling the lubricant up. Typically, the wetting ridge can be of the order of a few micrometers, distorting and hiding the contact line. This makes it very difficult to measure the true contact angle on the SLIPS and, instead, an apparent contact angle (differing from the contact angle on lotus leaf mimics) is used [70,71]. In the event of a drop moving on a liquid-infused surface, the wetting ridge also moves with the contact line, contributing significantly to the lubricant's depletion. Although the lubricant taken along with a moving drop is one of the most important mechanisms of lubricant loss, a fundamental understanding of the formation of wetting ridges is still lacking. For more information about the formation of wetting ridges and induced depletion, we refer to [72–74].

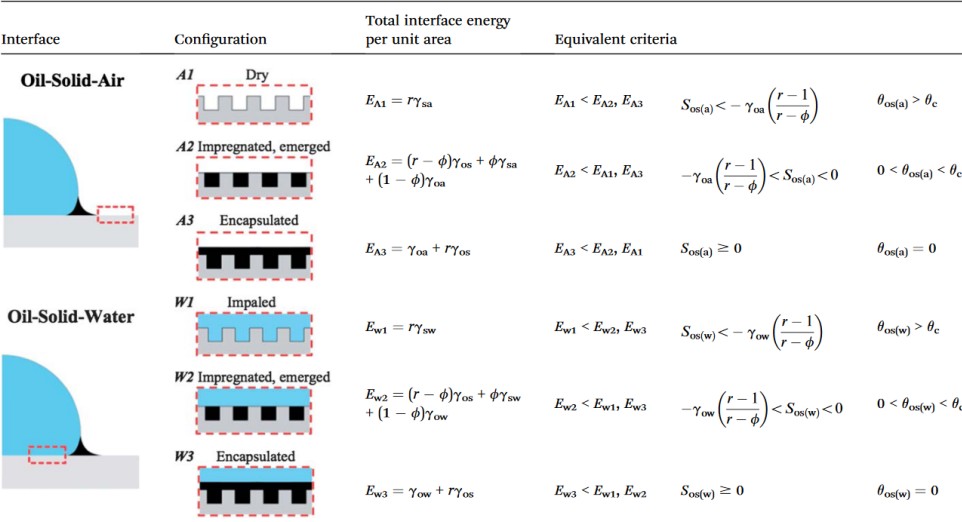

| Interface | Configuration | Total interface energy per unit area | Equivalent criteria | | | |
|---|---|---|---|---|---|---|
| **Oil-Solid-Air** | **A1** Dry | $E_{A1} = r\gamma_{sa}$ | $E_{A1} < E_{A2}, E_{A3}$ | $S_{os(a)} < -\gamma_{oa}\left(\dfrac{r-1}{r-\phi}\right)$ | $\theta_{os(a)} > \theta_c$ |
| | **A2** Impregnated, emerged | $E_{A2} = (r-\phi)\gamma_{os} + \phi\gamma_{sa} + (1-\phi)\gamma_{oa}$ | $E_{A2} < E_{A1}, E_{A3}$ | $-\gamma_{oa}\left(\dfrac{r-1}{r-\phi}\right) < S_{os(a)} < 0$ | $0 < \theta_{os(a)} < \theta_c$ |
| | **A3** Encapsulated | $E_{A3} = \gamma_{oa} + r\gamma_{os}$ | $E_{A3} < E_{A2}, E_{A1}$ | $S_{os(a)} \geq 0$ | $\theta_{os(a)} = 0$ |
| **Oil-Solid-Water** | **W1** Impaled | $E_{w1} = r\gamma_{sw}$ | $E_{w1} < E_{w2}, E_{w3}$ | $S_{os(w)} < -\gamma_{ow}\left(\dfrac{r-1}{r-\phi}\right)$ | $\theta_{os(w)} > \theta_c$ |
| | **W2** Impregnated, emerged | $E_{w2} = (r-\phi)\gamma_{os} + \phi\gamma_{sw} + (1-\phi)\gamma_{ow}$ | $E_{w2} < E_{w1}, E_{w3}$ | $-\gamma_{ow}\left(\dfrac{r-1}{r-\phi}\right) < S_{os(w)} < 0$ | $0 < \theta_{os(w)} < \theta_c$ |
| | **W3** Encapsulated | $E_{w3} = \gamma_{ow} + r\gamma_{os}$ | $E_{w3} < E_{w1}, E_{w2}$ | $S_{os(w)} \geq 0$ | $\theta_{os(w)} = 0$ |

**Figure 5.** Scheme showing the possible states on liquid-infused surfaces outside and underneath the drop in column 2. The corresponding total interfacial energies per unit area are given in column 3. The equivalent criteria for the stability of each configuration are given in column 4. Reproduced from Ref. [63] with permission from the Royal Society of Chemistry.

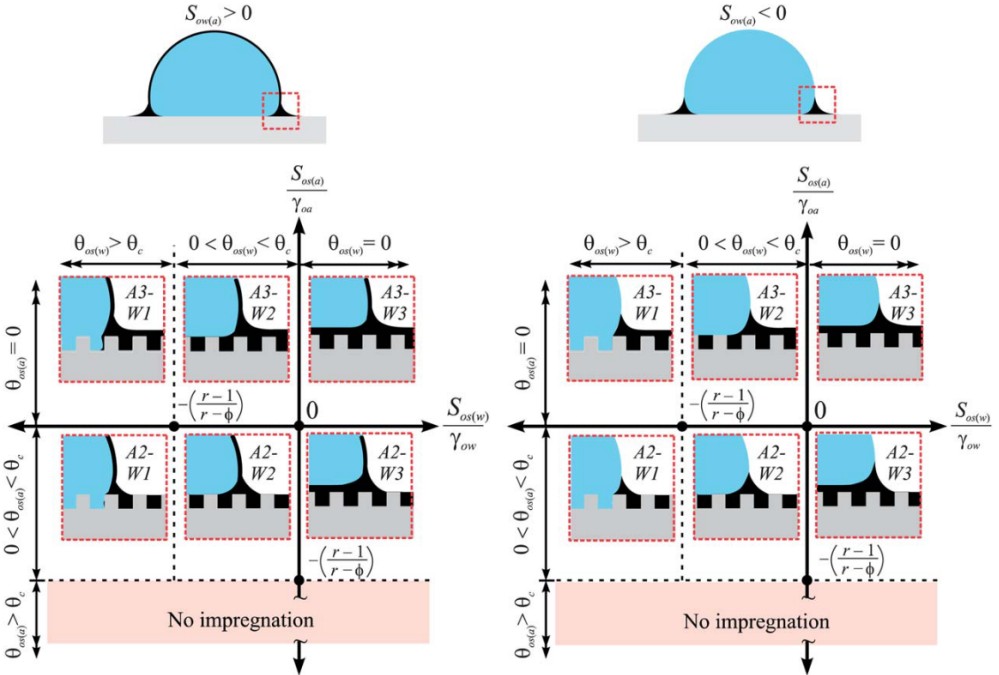

**Figure 6.** Schematic representation of the possible thermodynamic states of a droplet on a liquid-infused surface, as constructed by Smith et al. for both cases when $S_{ow}(a) > 0$ and when $S_{ow}(a) < 0$. Reproduced from Ref. [63] with permission from the Royal Society of Chemistry.

## 3. Durability of Slippery Liquid-Infused Porous Surfaces (SLIPS)

The durability of SLIPS, in diverse working environments, is a factor limiting their practical application [69]. The lubricant infused within the porous surface is what provides the slipperiness to SLIPS. The deterioration of SLIPS is primarily caused by a combination of oil depletion and the smoothing of the nanoporous functional layer [75]. A multitude of mechanisms can cause the erosion of the lubricant from the surface protrusions; we mention the most important ones below.

- Under high mechanical stress, there could be damage to the porous structure that is responsible for holding the lubricant, resulting in the leakage of the lubricant. Alternatively, the lubricant can also drain itself due to shear stress [76,77].
- The interaction of acidic or alkaline contaminants with porous substrates can result in corrosion, leading to damage to the porous structure. Reactions with the lubricant lead to chemical degradation through reactive reactions [78].
- High temperatures can cause the lubricant to evaporate or decompose, which reduces the slipperiness of SLIPS [79].
- Exposure to UV radiation can break chemical bonds, degrade molecular structures or cause oxidation reactions. This leads to changes in the surface properties and structural integrity of the materials [77,80].
- The formation of wetting ridges around the external droplets can lead to the depletion of the lubricant during droplet motion on the surface [73].
- Cloaking of the lubricant over the external working droplet is another mechanism that induces the loss of the lubricant [81,82].

The mechanical resilience of SLIPS plays a pivotal role in determining their durability: SLIPS must possess the ability to withstand rigorous mechanical stress. During the process of deicing, the application of shear stress can result in Rhebinder's fracture and the depletion of the lubricant [76]. When used as a power line wire coating to repel water and reduce the corona ignition voltage, SLIPS exhibited a high rate of infused oil depletion [80]. However, there are durability concerns regarding the practical application of SLIPS. The mechanical stability of SLIPS can be evaluated through established durability testing methods such as tape peeling, cross-cutting, the tangential abrasion durability test, the dynamic impact durability test, the liquid bath durability test and others [38,78].

The longevity of SLIPS can be tuned via a number of parameters, such as the chemical composition and physical characteristics of the lubricant layer, the roughness of the solid substrate and the prevailing environmental conditions [83]. Gaining comprehensive insights into the fundamental mechanisms of lubricant depletion and the obstacles linked with the durability of SLIPS is, thus, imperative in designing durable SLIPS. In the following, we will give a concise review of the studies that focus on increasing the durability of SLIPS via controlling some system properties.

### 3.1. Substrate Properties and Durability

The physical and chemical properties of the substrate play an important role in lubricant retention. In the following, we discuss studies based on exploiting the substrate properties to enhance the longevity of SLIPS.

### 3.1.1. Surface Structure

The size and nature of the pores, on the surface, are crucial in ensuring the secure anchoring of the lubricant within the substrate [84]. The correct choice of roughness can facilitate the stable retention of the lubricant via the combined effects of capillarity and van der Waals forces, which surpass external shearing forces [85,86]. The incorporation of a regular and periodic wettability pattern can significantly reduce the risk of oil drainage due to tensile and gravitational forces [87]. Furthermore, a substrate with a small pore size and high porosity exhibits excellent durability and anti-icing properties [88,89]. The small pore size creates high capillary pressure, effectively holding the lubricant inside the substrate and enhancing the durability. Additionally, the high porosity allows for a larger contact area between the contaminant liquid and the lubricant, leading to effective self-cleaning.

Several methods have been developed to create roughness or pores of desired sizes. Roughness can be generated by either adding a textured layer (bottom-top) or removing a portion of the surface (top-bottom) [69,90–93]. Subtractive methods include photolithography, soft lithography, laser ablation and etching techniques [43,47,94–100]. Additive techniques include methods such as deposition (chemical vapor deposition, electrochemical

deposition, liquid-phase deposition techniques), sol–gel methods, self-assembly and layer-by-layer assembly [101–107].

Thermal spray coating is a cost-effective and eco-friendly option for the fabrication of SLIPS substrates using specific spray parameters. Low-pressure cold spray and flame spray are preferred methods: while the cold spray is ideal for the on-site fabrication of thick, multi-layer polymer coatings on heat-sensitive substrates, the flame spray method is more suitable for creating high-porosity polymer coatings [108]. The most commonly used polymers are low-density polyethylene and polyether ether ketone. Flame-sprayed polyethylene coatings (FS-PE) show greater oil locking than cold-sprayed polyethylene (CS-PE) and polyether ether ketone (CS-PEEK) coatings due to their evenly distributed and closed-cell structures, which retain more lubricant under shear stress [108]. Yuanzhe et al. used a titania–polyurea ($TiO_2$–SPUA) spray coating to construct a regularly hydrophobic surface texture on a polyurea coating system [109]. The surface created by this method was found to inhibit biofilm growth and was also eco-friendly.

Anodization techniques, in which the electrolytic surface oxidation of aluminum or aluminum alloys is performed via a controlled applied voltage, allow the better texturing of porous structures. Surfaces created using this technique also show better shear, thermal and corrosion resistance [110–113]. Zhang et al. [114] developed a facile method for the fabrication of SLIPS using a polyHIPE (poly high internal phase emulsion) membrane as a porous substrate. PolyHIPE offers pore size control via crosslinker concentration during polymerization. The study concluded that polyHIPE's hierarchical porous structure exhibited self-healing against scratching damage and also enhanced the durability of SLIPS.

A number of studies have shown hierarchical structures to be more robust and possess better self-healing. Kim et al. studied surfaces with different roughness scales and reported that surfaces with uniform nanofeatures provided improved pressure stability and low contact angle hysteresis [115]. Hybrid porous (nano-/microstructure) surfaces were found to prevent algal attachment due to an unfavorable topography and stabilized surface composition, while the surfaces endowed with irregular nanoholes were reported to reduce bio-adhesion [116]. Combined micro- and nanostructures, when studied in seawater and anti-icing applications [108], exhibited greater stability than individual micro- or nanopore structures. This was attributed to the loss of attachment points due to the increased area of the lubricant film on the interface [116]. In case of lubricant depletion, residual infused lubricant that is encapsulated inside nanofeatures spreads over micropillars and helps in self-healing. In Figure 7, it can be seen how hierarchical structures can enhance the durability and self-healing of SLIPS. However, in vibration-induced depletion, integrating sub-micron features on micro-dimensional pillars leads to less resistance to lubricant loss compared to monoscale surface pillars [117].

Liu et al. [118] prepared SLIPS using a fluorinated hierarchically micro-/nanostructured silicone rubber surface. The surface was evaluated for its performance under stimulated rainfall tests, icing/deicing cycles and ice adhesion strength measurements. Hierarchical nanostructures produced higher capillary pressure within nanopores, which locked the lubricant more effectively within the nanostructures and led to a significant improvement in the durability of the surface. Boveri et al. [119] used the sol–gel method to fabricate SLIPS with alumina and silica nanoparticles, resulting in a hierarchical structure with an organic coating. Fluorinated Krytox oil was chosen as the lubricant for its high repellency and exceptional durability below the freezing point. The authors reported that silica-nanoparticle-based SLIPS were less durable compared to alumina-nanoparticle-based SLIPS in freeze/thaw cycles. These findings were attributed to the larger micronic pore size (1 μm) created by silica nanoparticles in comparison to the smaller cavities (50 nm) created by alumina. The authors explained the findings based on the Dzyaloshinskii–Lifshitz–Pitaevskii model. They concluded that stronger Van der Waals forces in the alumina-based SLIPS led to stronger capillarity and better self-recovery in comparison to the silica-based SLIPS [119].

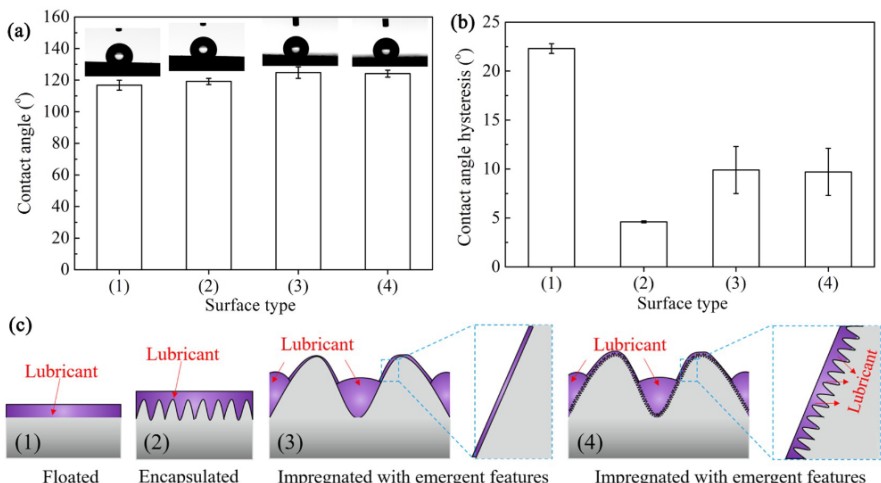

**Figure 7.** (**a**,**b**) show the contact angle and contact angle hysteresis, respectively, on the four different surfaces that are shown in (**c**). As can be seen, a hierarchical structure helps in both lubricant retention and re-distribution. Reproduced from [116] with permission from ELSEVIER.

In the same context, Tan et al. created hierarchical structures with micropyramidal holes and a porous nanostructure and called them P-SLIPS. P-SLIPS exhibited an increased contact area between the infused liquid and substrate due to the improved adsorption of inverted pyramid-shaped nanostructures. Robust micropyramidal sidewalls protect internal nanostructures and prevent liquid erosion [120]; a scheme depicting this is shown in Figure 8. Furthermore, networked surface structures have been reported to provide SLIPS with more robustness and durability [121,122]. Surfaces textured with interconnected microchannels and crosslinked nanosheets can firmly lock in and store liquid.

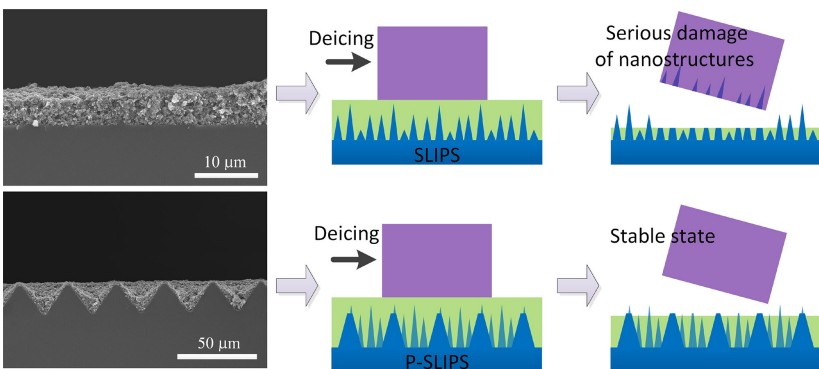

**Figure 8.** Scheme showing the SLIPS with micropyramidal holes providing better durability during deicing process. Reproduced from [123] with permission from ELSEVIER.

Lv et al. [124] developed a novel method for the fabrication of SLIPS substrates using etched honeycomb pores on aluminum sheets coated with 1-octadecanethiol solution. The increased surface roughness led to enhanced lubricant retention and durability. Electrochemical corrosion testing of the surface in saltwater showed superior durability, indicating the immense potential of honeycomb-shaped SLIPS in applications in harsh marine environments. Although the increased roughness enhances lubricant retention, it may also result in greater lubricant evaporation due to the larger surface area. Cai et al. [125,126] proposed a solution to this dilemma by using arrays of micropores with inclined walls, which minimize the rate of evaporation without compromising the high aspect ratio of the nano-/microstructure (see Figure 9 for the scheme). Inclined walls further enhance the Laplace pressure, increasing the durability under high shear stress compared to vertical walls. These surfaces also effectively mitigate biofouling on cavity surfaces under shear stress by increasing the micro-array aspect ratio. Dynamic biofouling tests have shown that

a 20-degree inclined micro-cavity can reduce bacterial surface coverage by up to 30% due to superior lubricant retention [125].

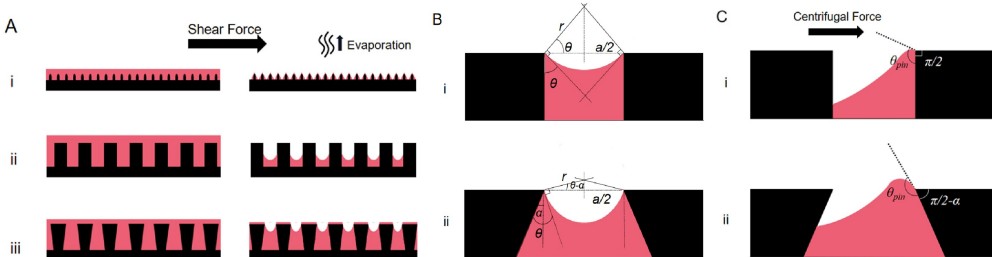

**Figure 9.** (**A**) shows how the lubricant profile changes on a lubricant-infused surface, under a shear force, for three different types of surface protrusions: (i) nanostructured substrate, (ii) substrate with a vertical microstructure and (iii) substrate with an inclined microstructure. (**B**) demonstrates the Laplace effect within the lubricant in two types of microcavities: (i) a vertical microcavity and (ii) an inclined microcavity. (**C**) shows lubricant pinning at the edge of a vertical microcavity. Reproduced from [125] with permission from ELSEVIER.

Another strategy to enhance the capillary forces with minimized evaporation is inspired by the oil-reserving microporous structures found in citrus fruits. Han et al. created pores with narrow opening and re-entrant structures on SLIPS substrates. Such pores strikingly reduced the lubricant loss by shear drainage due to the enclosed nature of the sidewalls. The re-entrant architecture of micro-citrus structures allows over hundreds of water droplets to slide on the surface, which is one hundred times more than vertical micropillar structures [127]. Another means of enhancing the lubricant retention is to use substrate coatings. More about surface coating is being discussed in Section 3.1.2. Liquid-infused nanostructured surfaces created by Xiao et al. exhibited exceptional liquid retention due to the coordinated action of a branched nanotree structure and the strong fixation of infused dimethyl-silicone to zinc oxide (ZnO) nanorod branches. This fixation is facilitated by Zn-O-Si bonds formed through photocatalytic reactions under UV light [128]. Further, Lv et al. suggest that substrate coating techniques can be used to replace harmful fluorinated lubricants or expensive siloxane with less harmful lubricants. Microscopic petal-like structures on aluminum sheets can replace fluorinated lubricants. The process involves immersing an aluminium (Al) substrate in a zinc-containing solution, followed by heating, chemical modification and infusion with a less harmful silicone oil. The resulting surface with an Al-ZnO petal-like structure [129,130] exhibits superior oil-locking ability and exceptional robustness under various tests and represents a sustainable fluorine-free alternative [129] that can effectively reduce the production costs of SLIPS surfaces [129].

SLIPS technology reduces resource wastage by preventing the adhesion of viscous substances in storage. However, during transportation and supply, the surfaces are exposed to intense vibration and shear stress. To improve the durability, minutely textured stainless steel substrates coated with thermoplastic polymer sheets have been introduced. These surfaces use two topographical profiles: laser-induced periodic surface structures (LIPSS) and multiscale structures (MS). MS provide more accurate replication but lose more lubricant due to reduced capillary pressure, whereas LIPSS exhibit negligible lubricant loss and superior shear durability, albeit varying with liquid viscosity. Immersion tests show that LIPSS's highly regular topography locks in an ultrafine layer of lubricant, suppressing leaching [117].

Laney and colleagues [131] conducted a thorough investigation into the loss of lubricant on SLIPS that were composed of three distinct ordered nanostructures, namely nanotubes, nanoholes and nanopillars, as shown in Figure 10. Droplets of an external fluid were continuously deposited on the SLIPS created using the above three nanostructures, resulting in depletion of the lubricant due to the formation of a wetting ridge. The morphology of each nanostructure was then analyzed further, to understand their lubricant retention mechanisms. Nanopillars have an open structure that facilitates easy redistribu-

tion of the lubricant, allowing for lubricant recovery until depletion becomes significant. Nanoholes retain the lubricant due to the capillary forces above the pores; however, the thin layer of lubricant covering the large solid fraction is susceptible to displacement along with droplets. In addition, the absence of an interconnected structure hinders lubricant recovery. Remarkably, nanotubes possess the attributes of both nanopillars and nanoholes, demonstrating the best lubricant retention [131]. The open structure of nanotubes enables the redistribution of the lubricant to depleted areas. Furthermore, the lubricant reservoir within the pore and the low solid fraction minimize the impact of the exposed solid structure on droplet mobility. This leads to a reduction in the amount of lubricant dragged alongside the wetting ridge due to a decrease in the time that droplets spend on the surface. Although the surface topography plays a role, the chemical properties of the surface have a profound effect on lubricant adhesion within the surface protrusions. In the following, we will briefly discuss the effects of surface coatings on lubricant retention.

### 3.1.2. Substrate Coating and Durability

To successfully achieve the wetting of a substrate with a lubricant rather than the working droplet, it is crucial to increase the chemical affinity between the lubricant and the substrate [63]. A promising method to enhance the chemical affinity involves reducing the surface energy of the substrate through the application of a hydrophobic coating, which effectively stabilizes the lubricant layer on the surface [132]. Alternatively, modifying the substrate with functional groups that reduce the surface energy or adding adhesive groups can promote the stabilization of the lubricant layer on the substrate underneath.

The stabilization of the lubricant layer has traditionally been achieved through the functionalization of the substrate via silanization. This process involves the removal of hydroxyl groups (-OH) from the substrate through the use of silane, resulting in the formation of covalent bonds between silicon and the substrate via Si-O-M bonding (where M represents the substrate) [68,133]. Subsequently, an adhesive group is added. Aizenberg et al. demonstrated that a porous substrate fabricated via the layer-by-layer self-assembly of silica nanoparticles and coated with fluorinated silica can stabilize the fluorinated liquid [101].

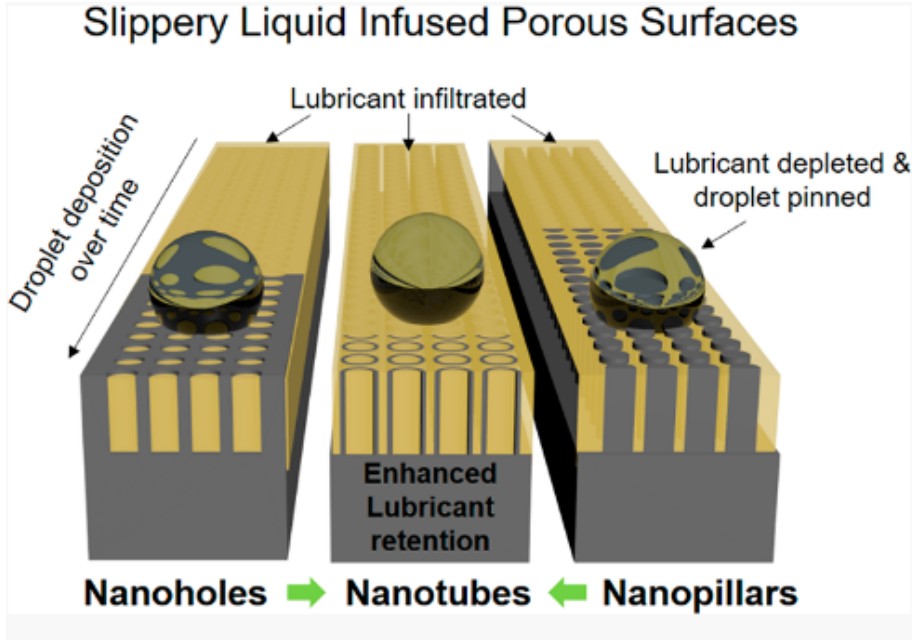

**Figure 10.** Schemetic representation of SLIPS with nanoholes, nanotubes and nanopillars. Nanotubes show the attributes possessed by both nanoholes and nanopillars. Reproduced from [131] with permission from the American Chemical Society.

Cheng et al. employed the grafting of polydimethylsiloxane (PDMS) brushes on platinum to achieve liquid-like behavior [134]. Zhang et al. fabricated a liquid-infused surface by functionalizing the substrate with 1,3,5,7-tetramethylcyclotetrasiloxane and then adding a layer of divinyl-terminated PDMS, which immobilized the silicone oil [135]. Jiang et al. [136] chemically grafted polydimethylsiloxane (PDMS) onto photocatalytic ZnO nanorods by Zn-O-Si covalent bond formation. Chemical grafting occurred when the siloxane bonds of the PDMS molecules were cleaved by the activated hydroxyl groups activated by ZnO nanorods under UV exposure, followed by covalent bond formation between the siloxane-based chains of PDMS and ZnO nanorods. The infused non-bound silicone oil was strongly locked into this grafted polydimethylsilane layer because of the strong intermolecular forces between the grafted layer and the lubricant, yielding a thermally, chemically and mechanically robust SLIPS with no harmful after-effects [137]. Another approach is to graft a long polymer chain onto the substrate, which can crosslink with the lubricant and stabilize it within the porous structure [138,139]. For instance, Coady et al. utilized a UV-cured polymer resin to crosslink the substrate with the lubricant, effectively enhancing the durability of the SLIPS [140].

The addition of nanoparticles has also been used as a method to functionalize substrates with hydrophobic coatings [141,142]. Adding a binder to the substrate can stabilize the lubricant to the substrate roughness. For example, Li et al. [141] fabricated a polymeric crosslinked network via the hydrolysis of three types of silanes, and then coated the networked structure with $SiO_2$ nanosol. They further added a UV-curing adhesive to the substrate to make it hydrophobic, and after exposure to UV radiation, this network could firmly hold the lubricant.

Zhu et al. [143] devised a novel approach for the synthesis of hydrophobic nanoparticles, utilizing dual nanosized $SiO_2$ that were modified with low surface energy groups, namely hexadecyltrimethoxysilane (HDTMS) and stearic acid (SA). The nanoparticles are known as $H-SiO_2$ and $S-SiO_2$ nanoparticles, respectively. The modified nanoparticles were employed to coat two aluminum substrates that were previously treated with an epoxy resin as a binder. The coating of $H-SiO_2$ and $S-SiO_2$ nanoparticles on the respective substrates, followed by infusion with silicone oil and camellia oil, resulted in the creation of SLIPS with improved lubricant retention capabilities. The modified $SiO_2$ nanoparticles successfully functionalized the rough structure of the substrates, and the epoxy resin layer demonstrated strong bonding with the modified nanoparticles and the substrate.

In a study by Long et al. [144], attapulgite (APT) was employed as a substrate with a pre-existing micro-/nonporous structure that harbored reactive hydroxyl (-OH) groups, which were subsequently functionalized by octadecyltrimethoxysilane (OTMS). The binder used in this approach was aluminum phosphate (AP), which was incorporated into the APT-OTMS-AP suspension. The suspension was then sprayed onto a magnesium (Mg) substrate, resulting in a SLIPS that demonstrated the remarkable stability of the infused silicone oil. The use of APT as the substrate allowed for the creation of a robust and durable SLIPS due to its unique surface morphology and reactivity toward functionalization. Another innovative technique is to employ $\pi$-OH and $\pi$-COOH interactions between the substrate and the lubricant, which can stabilize the lubricant layer even on flat surfaces or hydrophilic surfaces [68,145,146].

Another important aspect of chemically modifying a surface is selecting the appropriate lubricant and substrate modifier for optimal compatibility. In a recent study by Xiang et al. [147], the authors chose six combinations of bare porous anodic aluminum oxide substrates, coated with different modifiers: linear alkane (n-octadecyltrimethosilane, OTS) and fluorinated alkane (1H,1H,2H,2H-perfluorodecyltriethoxysilane, FAS). These porous substrates were then infused with either silicone oil (SO) or perfluoropolyether oil (PFPE), as shown in Figure 11 [147]. The experimental findings revealed that while SO is able to fully wet an OTS-coated substrate, it is incapable of achieving the same level of wetting with an FAS-coated substrate. The reason for this difference is that silicone oil has a siloxane bond (-Si-O-Si-) attached to the methyl group (-CH$_3$), while PFPE is terminated by

a fluorine group (-CF$_3$). Since the fluorine group is more polar than the methyl group due to its high electronegativity, PFPE is more polar to silicone oil. In the case of the modifier linear alkane grafted on the substrate, it contains (-CH$_3$) groups, and fluorinated alkane contains (-CF$_3$) groups. Due to the same end group and polarity, silicon has higher chemical affinity and is more compatible when stabilized with linear alkanes (OTS) rather than FAS, and, similarly, PFPE has greater affinity when stabilized with FAS. Apart from the surface properties, the properties of the lubricant also play an important role in lubricant retention. In the next section, we will discuss these in detail.

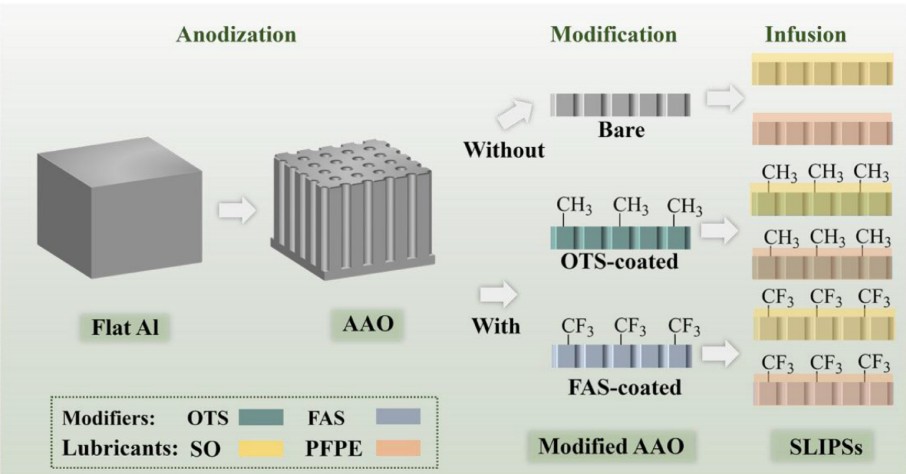

**Figure 11.** Scheme showing the fabrication steps of SLIPS unmodified, modified with OTS and modified with FAS coatings. Reproduced from [147] with permission from ELSEVIER.

### 3.2. Lubricant and the Durability of SLIPS

Although there are numerous lubricants available that may meet the selection criteria for the fabrication of SLIPS, it is important to choose a lubricant that satisfies specific requirements to ensure that the SLIPS remains stable and long-lasting [68]. In order to achieve this, the chosen lubricant must adhere to the following guidelines.

1.  Low surface tension is important for a lubricant used on slippery liquid-infused surfaces, as it allows the lubricant to spread easily and thoroughly over the surface. This property ensures that the lubricant can reach all areas of the surface and provide complete coverage, achieving effective lubrication.
2.  Lubricants used should be less volatile; a low vapor pressure is desirable because it means that the lubricant is less likely to evaporate, and thus less likely to become depleted over time, leading to high thermal durability.
3.  The lubricant must be chemically inert so that its slippery property is not diminished when reacting with the other contaminants.
4.  Cloaking and wetting ridges are the primary sources of lubricant depletion. The viscosity of the lubricant should be carefully chosen so as to reduce the loss of lubricant due to moving along with the droplet.

In the following, we will discuss the types of oil that are being used as lubricants, with their advantages and disadvantages. We will also discuss the effect of viscosity on lubricant depletion, which is an important aspect when choosing an appropriate lubricant.

### 3.2.1. Types of Lubricants

The lubricants most frequently documented in the literature include silicone oils, fluorocarbon oils, ionic liquids and edible oils.

**Silicone oils**—most commonly polydimethylsiloxane (PDMS)—consist of extensive chains of repeating silicon–oxygen (-Si-O-) units (see Figure 12). These lubricants have low surface tension, typically below 209 J/cm$^2$, which makes them good candidates for lubricants. In addition to being economical, silicone oils are non-toxic, and their low vapor

pressure confers high resistance to volatilization (approximately $10^{-1}$, kPas). Furthermore, they are chemically stable. However, due to their low surface tension, silicone oils can exhibit a tendency to flow out of materials. Moreover, they have limited miscibility with most working oils, with the notable exception of water [3,69,79,148–151].

$$CF_3\text{-}O\text{-}(\overset{\overset{\displaystyle CF_3}{|}}{\underset{\underset{\displaystyle F}{|}}{C}}\text{-}CF_2\text{-}O)_m\text{-}(CF_2\text{-}O)_n CF_3$$

PFPE, Krytox

silicone

**Figure 12.** Representative image of the structure of a fluorocarbon oil and silicone oil.

**Fluorocarbon oils** contain long chains of repeating carbon and fluorine atoms bonded together in a (C-F) configuration (see Figure 12). Due to the high binding energy of the (C-F) bond, these lubricants are thermally stable and resistant to degradation. However, the vapor pressure of fluorinated oils is relatively high, approximately $10^{-8}$ kPas.

The most commonly used fluorocarbon oils are perfluoropolyether (PFPE) and PDMS oil. The literature describes the Krytox series of fluorinated oils, which exhibit a wide range of viscosity. Despite their advantages, fluorocarbon oils can be costly and contain toxic fluorine, which makes them less environmentally friendly than other lubricants [41,69,115,150,152,153].

**Vegetable oils** are obtained from the seeds or fruits of plants, such as corn, linseed, peanut, cottonseed, coconut and palm. It is worth noting that these oils contain large fatty acid chains consisting of 12–33 carbon atoms, imparting excellent wear resistance. In addition, vegetable oils are non-toxic and readily available for extraction. However, vegetable oils are susceptible to reaction with contaminants, resulting in the loss of their lubricating properties. Additionally, they tend to have the highest vapor pressure, making them relatively less thermally stable than other lubricants [66,150,154,155].

**Ionic liquids** have gained increasing attention in recent years due to their exceptional thermal resistance [139]. These liquids are essentially molten salts at room temperature, composed of bulky and organic cations and weakly coordinating organic or inorganic anions of very low symmetry [156]. The most commonly used cations are imidazolium, pyridinium, ammonium and phosphonium. The anions are halide, tetrafluoroborate and nitrate and are attached with alkyl groups [157].

One of the unique features of ionic liquids is their extremely low vapor pressure, which contributes to their high thermal stability and wear resistance. They are also considered to be environmentally friendly. However, there are limitations to their use due their relatively high costs of production and their miscibility with polar droplets. Despite these limitations, research on ionic liquids continues to grow, as their distinctive properties hold promise for a wide range of applications [139,150,158–160].

Despite the potential environmental hazards and high costs of fluorinated oils, they are widely used in the literature as lubricants in SLIPS due to their advantageous properties, such as a wide range of viscosity and superior durability compared to other oils. However, Inoue et al. [161] have introduced an alternative solution with the development of fluorine-free SLIPS (F-free SLIPS). These F-free SLIPS are composed of a porous alumina substrate coated with hydrophobic alkyl phosphonic acid and infused with silicone oil, exhibiting greater durability than their fluorinated counterparts. This discovery provides hope for the identification of alternative oils to fluorinated oils for use in SLIPS technology.

To enhance the durability of SLIPS under defrosting conditions, researchers have turned their attention to thermo-responsive phase-transformable lubricants. These lubricants remain in a liquid state at normal temperatures, but undergo a phase transition to a solid state at low temperatures. In a recent study, Wang et al. [162] employed peanut oil as a phase-transformable lubricant and infused it into a polydimethylsiloxane (PDMS) substrate to fabricate SLIPS. The results demonstrated that the phase-transformable SLIPS achieved greater icephobicity and enhanced durability due to the solid lubricant being

strongly interlocked in the pores (Figure 13). The presence of the solid lubricant further improved the resistance against mechanical scratches and the stresses of icing and deicing. Moreover, researchers have also used lubricants that are solid at room temperature but turn into a liquid at higher temperatures (50–57 °C). Paraffin is the most commonly used lubricant of this type [163–165]. By utilizing such a lubricant, the loss of lubricant can be minimized, and the stability and durability can be enhanced.

### 3.2.2. Effect of the Viscosity of the Lubricant

The viscosity of lubricants is critical to the longevity of slippery liquid-infused porous surfaces. We will present here some of the research conducted to evaluate the viscosity-dependent durability.

Tonelli et al. conducted a study to investigate the impact of lubricant viscosity on the infusion and durability of slippery liquid-infused porous surfaces (SLIPS) with high porosity and complex roughness, fabricated using Portland cement [166]. The results of the study revealed that high-viscosity silicone oil, with viscosity of 20 kcst and 10 kcst, demonstrated superior performance, exhibiting minimal sliding angles (<4°) and lasting up to 4 and 2 weeks, respectively. In contrast, SLIPS infused with lower-viscosity oils (10, 350, and 1000 cst) experienced water droplets pinned to the surface shortly after infusion, with pinning observed within 1, 3 and 7 days, respectively. This can be attributed to the fact that low-viscosity liquids are readily adsorbed by capillary forces, resulting in easy infusion into the porous structure, but become prone to rapid drainage from the pores as well.

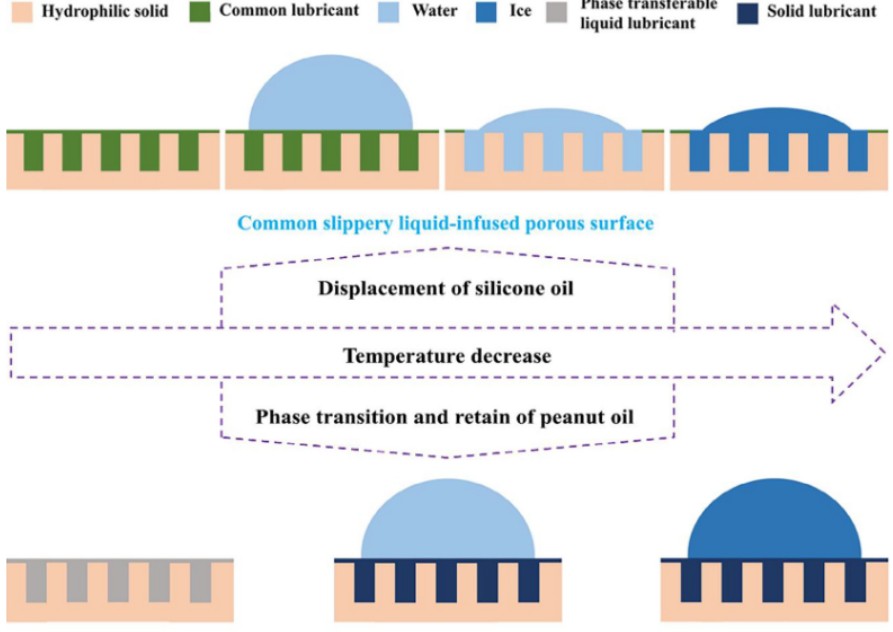

**Figure 13.** Scheme illustrating the behavior of SLIPS that are fabricated using hydrophilic substrates infused with silicone oil (**upper** image) and peanut oil (**lower** image) during the freezing process. Reproduced from [162] with permission from ELSEVIER.

Bandyopadhyay et al. investigated the impact of lubricant viscosity on the high-temperature durability of slippery liquid-infused surfaces on copper substrates using silicone oils of three different viscosities: 100 cst (*S1*), 350 cst (*S2*) and 1000 cst (*S3*) [167]. The authors found that high-viscosity lubricants exhibited a decrease in slipperiness at higher temperatures ($T_c$) compared to low-viscosity lubricants; reported values of $T_c$ were 300 °C, 350 °C and 430 °C for *S1*, *S2* and *S3*, respectively. This decrease is attributed to the reduction in lubricant viscosity at elevated temperatures, which leads to an increased

drainage rate through cloaking. Thus, increasing the viscosity will increase the durability of the SLIPS even at high temperatures.

Veronesi et al. conducted a study to investigate the impact of lubricant viscosity on the durability of SLIPS against chemical aging and mechanical abrasion. The authors observed that lubricants with high viscosity retained amphiphobicity for a longer duration when exposed to chemical degradation. Conversely, low-viscosity perfluorodecalin (PDF) lubricants containing short macromolecule chains exhibited greater susceptibility to degradation under abrasion [78]. Stoddard et al. experimentally showed that high-surface-tension lubricants were more stable in heated water immersion tests and high-viscosity liquids were most stable in the dynamic conditions of water jet impingement [168].

Moreover, in the context of anti-icing applications, the viscosity of lubricants plays a key role in determining the durability of SLIPS [169,170]. Research has shown that as the viscosity of the lubricant increases, there is an increase in contact angle hysteresis and a decrease in the sliding velocity of ice. These findings imply that higher-viscosity lubricants lead to an increase in ice adhesion strength and a decrease in anti-icing properties. The reduction in slipperiness or anti-icing properties with increasing viscocity occurs because the sliding droplet deforms the slippery oil layer, and more viscous oils are harder to deform. This causes a viscous drag on the sliding water and reduces the slipperiness of the surface. Lower-viscosity lubricants, on the other hand, can be easily separated. The viscosity of the lubricant also affects the self-healing properties of the SLIPS. Low-viscosity oils enhance the self-healing properties of SLIPS due to easy spreading, while SLIPS with high-viscosity lubricants show slower self-healing [171]. Thus, a good compromise regarding the viscosity of the oil is needed to achieve good durability as well as easy self-cleaning and self-healing at the same time. In the literature, [169], 200 cst silicone oil is preferred as it gives a sliding angle that does not exceed 5° and ice adhesion that does not exceed 10 kPa.

Sasidharanpillai et al. experimentally demonstrated that high-viscosity silicone oils (3000 cst and 5000 cst) exhibit lower lubricant loss due to cloaking and wetting ridges than low-viscosity oils (100 cst, 500 cst and 1000 cst). Rapid oil feeding from adjacent areas (high self-healing speed) causes more oil loss in low-viscosity SLIPS, as confirmed by dynamic wicking experiments (Figures 14 and 15). To achieve optimal durability and slipperiness, a viscosity range of 1000 cst to 3000 cst is recommended [172].

Cloaking and the formation of wetting ridges are two important mechanisms that result in lubricant depletion. With increasing viscosity, however, the occurrence of both cloaking [63,173,174] and ridge formation [73] decreases. This happens because it becomes more difficult to deform the liquid with higher viscosity. Hoque et al. [65] reported that, during steam condensation, the use of SLIPS as a condenser can result in lubricant loss due to condensate cloaking shedding. However, the utilization of a high-viscosity oil (5216 mPa) has been shown to markedly increase the lifespan of the slippery surface, extending it from a period of one month to eight months. This improvement is attributed to the high-viscosity lubricant's ability to reduce the departure speed of the lubricant during ridge formation and cloaking.

Considerable research efforts have been dedicated to investigating the relationship between the viscosity and durability of SLIPS. The results consistently indicate that fluids of higher viscosity offer superior durability under varying degradation conditions, including mechanical stresses and thermal and chemical degradation. However, a low-viscosity fluid provides specific benefits, such as improved slipperiness, anti-icing properties and ease of infusion, which are critical characteristics of SLIPS. Therefore, achieving an optimal balance between durability and slipperiness requires maintaining a moderate viscosity range. Despite the crucial role that viscosity plays in the development of SLIPS, there is a notable gap in research focused on identifying the optimal viscosity range. Further research in this area could significantly enhance the understanding of the complex relationship between viscosity and the properties of SLIPS, providing a foundation for the development of more advanced and effective SLIPS formulations.

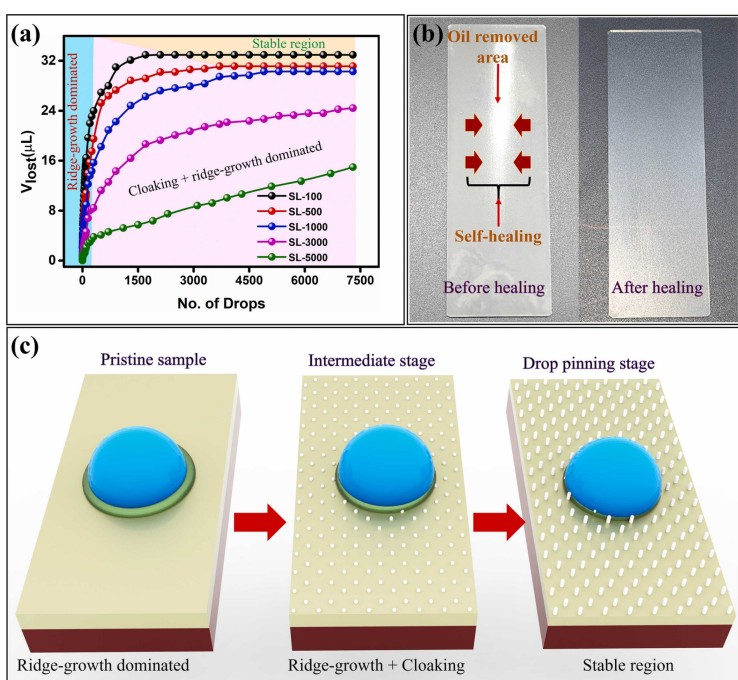

**Figure 14.** (**a**) shows the lost volume of lubricant as a function of the number of drops for different viscosities. (**b**) illustrates the self-healing of SLIPS and (**c**) shows how the wetting ridge expands as the surface goes through various stages of lubricant depletion. Reproduced from [172] with permission from ELSEVIER.

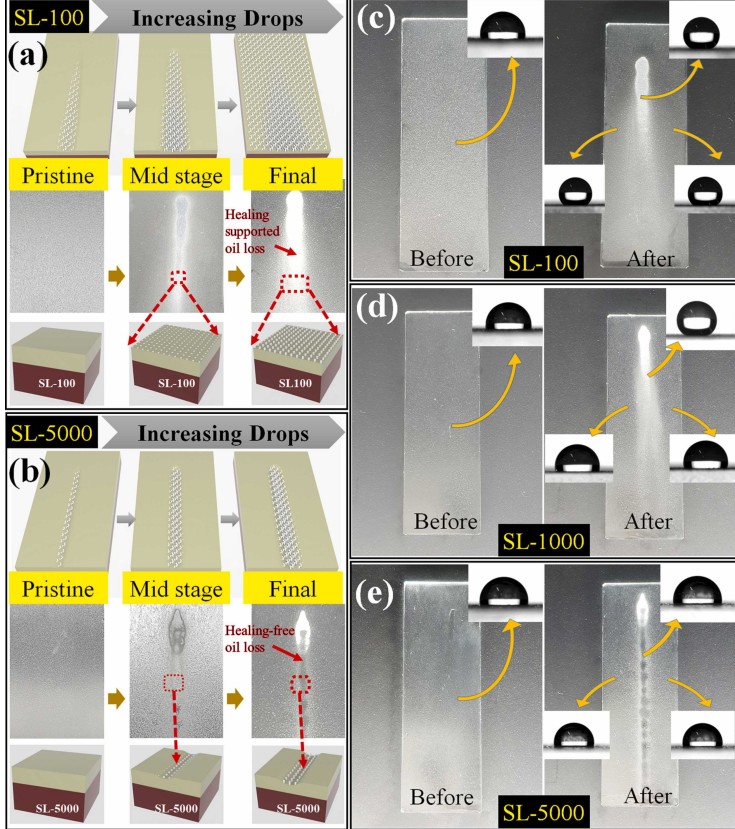

**Figure 15.** (**a**) is a schematic and (**b**) is a photographic depiction of oil loss with increases in drops for SL-100, SL-1000 and SL-5000. (**c–e**) show images of the SLIPS before and after 4500 drops are included. The contact angles are shown in the inset. Reproduced from [172] with permission from ELSEVIER.

### 3.3. Replenishment of the Lubricant

Despite ongoing efforts to increase lubricant retention, the depletion of the lubricant, so far, remains an inevitable occurrence that takes place over multiple application cycles. As an alternative, replenishment of the lost lubricant has been proposed to ensure the continued, efficient and effective operation of SLIPS. This, however, necessitates the availability of a lubricant reservoir; there can be replenishment either by an external reservoir or an internal reservoir.

#### 3.3.1. Replenishment through External Reservoir

To restore the depleted lubricant, various external sources can be utilized, such as spraying a lubricant from outside or dipping the depleted SLIPS into a lubricant-filled bath. However, these methods have some disadvantages that must be considered. Spraying a lubricant from outside lacks control, which can lead to an excess of lubricant on the surface and can potentially change the properties of the SLIPS [175,176].

An alternative approach to external reservoir replenishment is to use internal microchannel networks to deliver lubricant externally [177]. This can be accomplished by creating a 3D encased vascular system through molding [178] or a direct embedding vascular system (see Figure 16), which can provide external lubricant into the SLIPS via the vasculature [178,179]. Another method involves creating an internal channel within a polymeric matrix or solid that delivers lubricant through an inside-out mode from channels into a microporous PDMS matrix [177]. These internal methods provide better control over the replenishment process, which reduces the risk of overfilling the SLIPS and altering their properties.

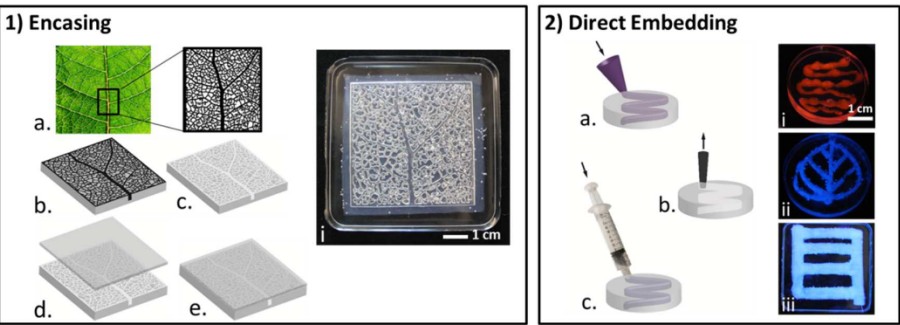

**Figure 16.** (1) shows a scheme for the creation of vascular networks (**i**) with encasement in PDMS via patterned molding, layer removal and sheet covering of another PDMS layer (**a–e**). (2) is a schematic representation of the embedded network creation process involving a 20 percent $w/w$ Pluronic *F*127 gel pattern embedded in uncured PDMS (**a**), followed by curing, cooling, liquid pluronic evacuation (**b**) and refilling (**c**) with silicone oil. The figure also shows images of sinusoidal (**i**), leaf-shaped (**ii**), and linear networks (**iii**) in PDMS with fluorescently dyed silicone oil. Reproduced from [178] with permission from the American Chemical Society.

Moreover, a novel continuous replenishment method for SLIPS was introduced by Seo et al. [180]. The method uses a brushed lubricant-impregnated surface, incorporating a brush with one end dipped in a lubricant reservoir and the other end attached to a rotating SLIPS, which is used as a condenser. The brush continuously provides oil to depleted regions via the capillary rise effect (see Figure 17 for the scheme). This approach ensures the consistent supply of lubricant to the SLIPS, making it an attractive option for continuous replenishment.

#### 3.3.2. Replenishment through Built-In Reservoir

Internal reservoir replenishment techniques can be employed to maintain the long-term performance of SLIPS without requiring an external lubricant supply. One such technique utilizes the self-healing properties of SLIPS to instantaneously replace the de-

pleted lubricant layer with liquid after any damage. This approach can be enhanced by increasing the efficiency of self-healing mechanisms as a means of lubricant replenishment [181–184].

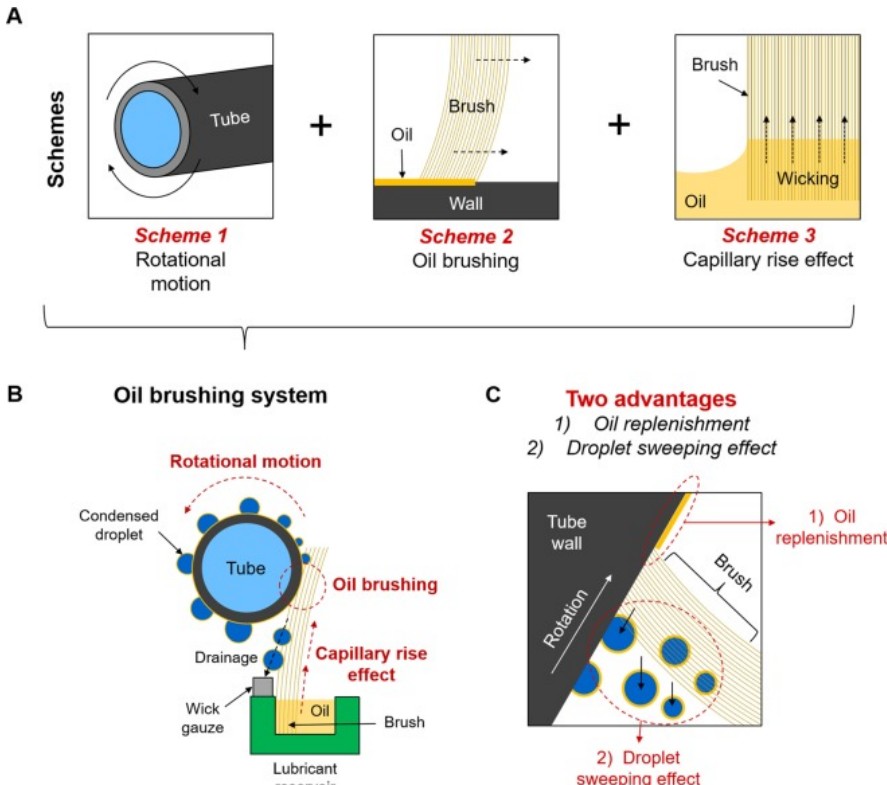

**Figure 17.** (**A**) is a schematic representation of the concept of replenishing via a brushing system, utilizing three schemes: the motion of the tube, oil brushing and the capillary rise effect. (**B**) is a depiction of an integral system that processes replenishment through a brushing system. (**C**) This system offers two distinct benefits, (1) efficient oil replenishment and (2) effective droplet sweeping, resulting in superior condensation heat transfer performance for a wide range of supersaturation levels. Reproduced from [180] under Creative Commons CC-BY license.

Stimuli-responsive SLIPS represent another effective method to replenish the lubricant. These SLIPS release lubricant in response to various stimuli, such as changes in temperature, electric field or light [185]. By utilizing stimuli-responsive SLIPS, lubricant release can be controlled to occur only when required. This affords the distinct benefit of preserving the internal lubricant reservoir, as its depletion rate is reduced due to the controlled release [186]. The advantages of these stimuli-responsive SLIPS include their ability to alter the wettability through external stimuli, and ongoing research is examining their potential benefits [185,187–189].

Moreover, lubricant-regenerable approaches can also be utilized to replenish SLIPS [141,190–193]. Zhou et al. were inspired by amphibians that continuously secrete mucous to maintain a hydrated body surface. They fabricated a skin-like SLIPS using a PDMS soft substrate, similar to the epidermis, and liquid paraffin to secrete a lubricant continuously [194].

Another promising approach involves the use of organogels for lubricant replenishment in SLIPS due to their ability to continuously secrete lubricant and exhibit good self-repairing properties [191,195]. These gels are composed of a liquid phase dispersed in a solid matrix, resulting in a soft material that can easily spread across surfaces. When used in SLIPS, organogels release the liquid continuously, which serves as a lubricant. This replenishment occurs through diffusion, and the amount of lubricant released can be controlled by altering the diffusion coefficient of the liquid phase. A more introductory

discussion of organogels will be presented in the next section. The major drawback to internal replenishment, however, lies in the fact that the internal reservoir of lubricant is not infinite and will become depleted after a certain period of time.

### 3.4. A Novel Liquid-Infused Surface: "Organogels"

A gel is a complex three-dimensional network composed of crosslinked polymer molecules that effectively trap a liquid within its structure. When the liquid trapped within the gel is composed of organic molecules, the resultant gel is categorized as an organogel. One of the unique characteristics of organogels is their solid-like consistency, which is a result of their crosslinked network structure. However, organogels also retain liquid-like properties due to the immobilized organic molecules.

There exist two principal methodologies for the production of organogels. Firstly, an organogel can be created by the swelling of a solid 3D molecular network structure with a liquid. The liquid interacts with the network structure in a manner that is balanced between the entropy of the polymer chain and the enthalpy of mixing. This equilibrium permits the liquid to be immobilized within the network, resulting in the formation of an organogel. Secondly, a one-pot gelation process can be utilized to generate organogels. This method involves the addition of a gelator to a liquid solvent. The gelator undergoes self-assembly or chemical polymerization, leading to the formation of a 3D network structure that is immersed in the liquid. This process also results in the formation of an organogel.

Organogel materials offer several advantages, such as self-healing, anti-sticking, anti-corrosion and anti-fouling properties. The gel formation phenomenon is thermo-reversible, meaning that reheating the gel will melt it, and the resulting solution can be transformed into a gel again by cooling; the heating–cooling cycle does not change the properties of the organogel [196,197]. Organogels exhibit self-healing properties, durability and good electrical properties [198]. Organogels, characterized by their liquid-infused slippery surfaces, are frequently synthesized by the swelling of silicone oil within a crosslinked polydimethylsiloxane (PDMS) polymer matrix [199]. In addition to PDMS, fluoro and alkyl polymers, such as polyvinylindene fluoride (PVDF) and polyethylene, have been examined as promising alternatives for organogel production [190,200,201]. Furthermore, the literature has extensively explored the use of less volatile alkanes as potential candidates for organogel synthesis, in addition to silicon and fluorine-containing oils [141,198,202–204].

Organogels have been previously studied but their ability to secrete a lubricant layer in anti-icing and anti-biofouling applications is novel [205,206]. According to Wang et al. [207], the use of organogels created using liquid paraffin and crosslinked polydimethylsiloxane resulted in a substantial decrease in ice adhesion strength for up to 35 icing–deicing cycles over a period of 100 days. The research conducted by Li et al. mimicked the epidermal glands of animals to fabricate an organogel with better self-healing than SLIPS. When a hybrid surfactant is introduced as a lubricant into a polydimethylsiloxane matrix, the lubricant is swallowed into the matrix, leading to the continuous excretion of oil, thus enhancing the self-repairing properties. The fabricated organogel exhibited remarkably low ice adhesion durability through 20 icing–deicing cycles and demonstrated exceptional thermal durability, with a negligible weight loss at a temperature of 100 °C for a duration of 60 h [141].

Kim et al. proposed a new method to produce durable ice-lubricating layers on colorful surfaces using porous silica aerogels instead of traditional elastomers [208]. This approach relies on incorporating excess silicone oil with the aerogel into PDMS to create a highly transparent and durable film. The aerogel's high lubricant absorption capacity, and its ability to disperse lubricant throughout the film, enables it to retain more lubricant while slowing down its release. This results in effectively reducing the ice adhesion strength and maintaining film stability for at least 20 icing–deicing cycles. This innovative approach shows great potential in creating long-lasting ice-lubricating layers on a variety of colorful surfaces.

Zhang et al. synthesized a slippery organogel by infusing silicone oil into a poly-dimethylsiloxane based polyurea (PDMS-PUa) matrix, which was prepared by reacting an $\alpha$, $\omega$-aminopropyl-terminated polydimethylsiloxane (APT-PDMS) with isophorone diisocyanate (IPDI) through the amine–isocyanate reaction [209]. The self-healing property of the resulting organogel was investigated by measuring the tensile force as a function of time at different temperatures, wherein the gel was cut into two pieces and then brought into contact. The authors reported that the organogel exhibited excellent self-healing properties, as is evidenced in Figure 18, which is reproduced from [209]. It was also reported that an increase in temperature accelerated the self-healing process. Moreover, the fabricated organogel demonstrated superior anti-biofouling properties under both static and dynamic conditions.

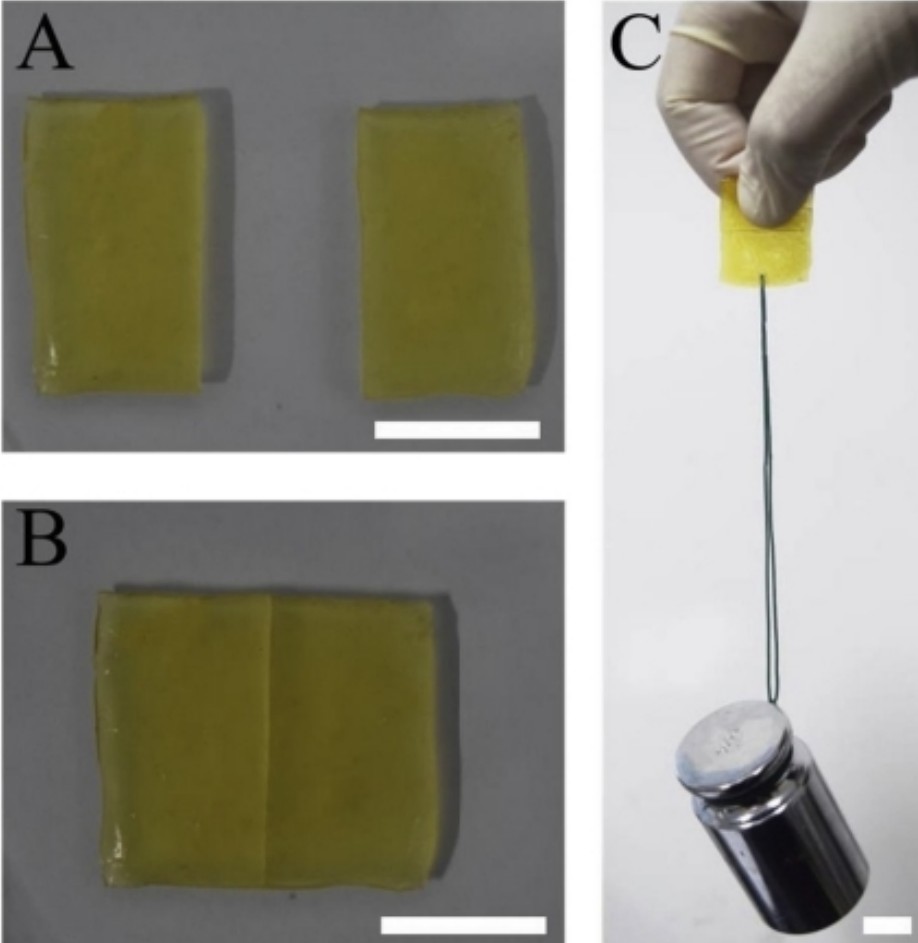

**Figure 18.** Photographs showing the self-healing ability of an organogel at room temperature (scale bar = 1 cm). (**A**,**B**) depict two samples placed in contact and their subsequent healing process, and (**C**) demonstrates the ability of the joined sample to withstand a 500 gm weight. Reprinted from [209] with permission from ELSEVIER.

Urata et al. designed organogels via the simple crosslinking of polydimethylsiloxane infused with an available commercial oil (as shown in Figure 19) [210]. By tuning the molecular weight of the infused oil and its content, three different transparent organogels were fabricated: (1) a non-syneresis organogel (NSG), (2) a self-lubricating organogel with high mobile oil (SLUG-1) and (3) a self-lubricating organogel with a highly viscous oil (SLUG-2). They reported that ice adhesion was zero in the NSG because of the continuous syneresis of the lubricant oil for more than 1 year.

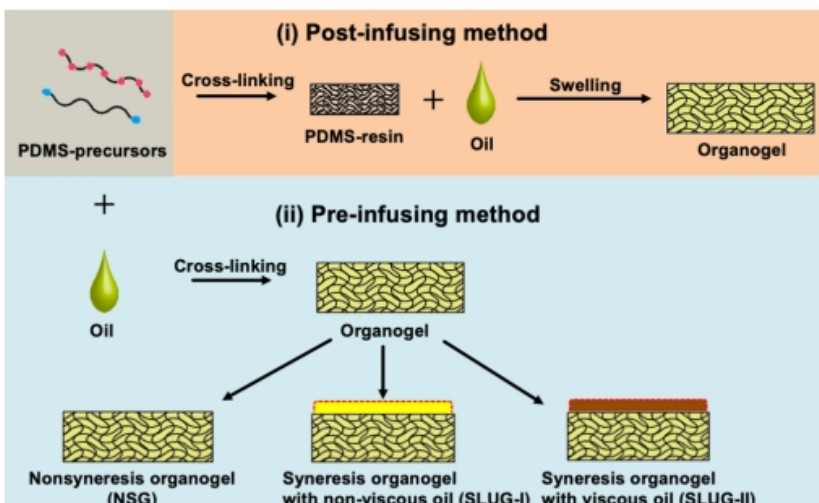

**Figure 19.** Scheme representing the two methods of fabrication of organogels by pre-infusion and post-infusion of oil [210]. Reproduced from [210] with permission from American Chemical Society.

Yu et al. used amphiphilic lubricants to create amphiphilic organogels (AmOG) composed of copolymer P (PDMS-r-PEG-r-GMA) and 2,2-diaminodiphenyldisulfide via an epoxy group addition reaction [211]. The resulting surfaces exhibited high stability and excellent anti-icing performance, delaying the freezing point of water by 1000 s and reducing the ice adhesion strength to 15.1 kPa. These novel AmOG offer a promising solution in developing long-lasting anti-icing materials. In the literature, many stimuli-responsive organogels, which offer better functionalities when used as SLIPS, have been reported [212,213,213,214]. Organogels have emerged as a revolutionary alternative to traditional SLIPS. Not only do they possess similar properties, but they also provide better stability and fabrication processes that are more straightforward. Organogels have a solid-like consistency and are self-healing, which makes them suitable for use in harsh environments. The immobilized liquid, in the organogel, offers unique features such as anti-biofouling, low friction and self-cleaning. Moreover, organogels demonstrate potential in the development of biomedical implants, where their non-stick properties can help to reduce bacterial adhesion and improve the biocompatibility of the implant [215]. Organogels can also aid in drug delivery systems, where they can prevent drug aggregation and enhance drug release kinetics. With their self-assembled networks of organic molecules that immobilize a liquid, organogels offer several advantages, including versatility, tunability and ease of synthesis. Overall, organogels are an emerging, highly promising alternative to traditional SLIPS and hold tremendous potential in the field of surface engineering.

## 4. Summary and Future Prospects

Liquid-infused surfaces have demonstrated immense potential for the design of anti-fouling surfaces that can resist unwanted accumulation when placed in contact with aqueous media, organic fluids or biological organisms. These surfaces are created by infusing a low-surface-tension liquid into the protrusions of a textured solid surface. This arrangement creates a chemically homogeneous and atomically smooth interface. SLIPS are capable of repelling a variety of liquids, exhibit pressure stability and are able to self-heal against any mechanical damage. Moreover, SLIPS show essentially no contact line pinning of external liquid, which makes it very easy for these surfaces to self-clean. These excellent properties, however, rely on the presence of sufficient lubricant within the surface protrusions. The lubricant, being a liquid, is susceptible to depletion due to a number of mechanisms, such as evaporation, gravitational drainage and shear (see Section 3 for more details). Retaining the lubricant is one of the most important criteria for the long-term application of SLIPS in various environments. This can be only achieved by developing

a microscopic understanding of the wetting behavior of SLIPS as a function of various system parameters.

Thus far, most of the studies regarding liquid-infused surfaces either focus on demonstrating their potential in various applications [9,44,48,93,216–220] or focus on the novel methods of fabrication. Although the methods of fabrication of SLIPS are beyond the scope of this review, we mention here some articles that interested readers could refer to [9,91,168,221]. Systematic studies of the effect of various system parameters on the wetting behavior of SLIPS, comparatively, remain less reported [63,65,81,86,88,89,169,222–225]. A detailed understanding of the mechanisms and pathways of lubricant depletion is fundamental to enhance the durability of the SLIPS. Well-matched solid and liquid surface energies, the correct size of surface protrusions, optimal lubricant viscosity and the chemical properties of both the liquid and solid play an important role. A comprehensive understanding of the effects of these parameters on the wetting behavior and drop dynamics is essential for the development of SLIPS that can be commercially used. Theoretical and computational tools can be very useful here; however, so far, such studies have been sparse [67,70,70,71,89,225–230].

The theoretical understanding of SLIPS needs to be developed further. An understanding of how the transition from an encapsulated/impregnated to impaled state takes place as a function of various physiochemical parameters, and how the wetting ridges are formed, is crucial in designing robust SLIPS. Understanding the effect of the system parameters on the drop dynamics, and how the ridges move with the drop, is essential in reducing lubricant depletion. These issues should be addressed using theoretical/computational tools. Such studies, when done systematically, would be vital in providing optimal design parameters for durable SLIPS.

As we have discussed in Section 3.1.1, surface topography plays an important role in keeping the lubricant in place. The size of the pores and the porosity of the surface are some of the control parameters to tune the wetting behavior of SLIPS. Smaller (nanosized) pores and greater porosity have led to better lubricant retention due to increased capillarity. Hierarchical structures are reported to perform even better when it comes to enhanced durability and self-healing. One, however, has to find a suitable compromise between increased capillarity and increased evaporation; pores with tilted walls have been proposed to achieve this balance. Increasing the affinity between the substrate and the lubricant by using some surface modifier, or by using polymers that act as crosslinkers between the lubricant and the surface, is another significant approach toward increasing lubricant retention. More information about increasing the attraction between the substrate and the lubricant can be found in Section 3.1.2. Possibly, a combination of more than one strategy could lead to better lubricant retention.

Another important aspect in designing a SLIPS is the choice of the lubricant. The physical and chemical properties of the lubricant play an important role as well. We have discussed the types of oils and liquids that have been used so far as lubricants, with their advantages and disadvantages, in Section 3.2.1. The viscosity of the lubricant plays a vital role in the lubricant retention and self-healing abilities of the SLIPS. As discussed in Section 3.2.2, a suitable choice of viscosity is needed in order to ensure easy infiltration and decreased lubricant loss while still maintaining the self-healing attributes. Designing SLIPS that could respond to some external fields, such as temperature, mechanical stimuli or electric/magnetic fields, can offer additional flexibility and better control over the wetting behavior and droplet dynamics of the SLIPS [185]. External stimuli have emerged as a promising means of minimizing lubricant loss, as well as a means of replenishment of the depleted lubricant. Replenishment of the lubricant has emerged as a useful strategy to compensate for the lubricant lost. This inside-out infused approach self-regulates the replenishment of the lubricant and, thus, maintains the SLIPS' functionalities. The complicated fabrication process and the vulnerable surfaces, however, are challenges that have to be addressed beforehand [74]. We have discussed the techniques for replenishment and their advantages and disadvantages in Section 3.3. We have also briefly discussed organogels in Section 3.4; these gels offer superior self-healing and anti-fouling in compari-

son to the traditional SLIPS. The use of organogels in place of conventional SLIPS, however, needs more research as the concept is rather new.

To conclude, the subject of SLIPS has drawn wide attention from researchers in the last decade. Although there has been considerable progress made by researchers, the commercial use of SLIPS still remains a goal to achieve. The retention of the lubricant within the surface protrusions is still a challenge. Creating functionalized surfaces and using external stimuli are some of the most promising approaches toward increasing the durability of SLIPS. Moreover, combining more than one of the proposed strategies could also be helpful. We believe that developing a better theoretical understanding of the SLIPS, and using simulations to achieve the optimized system parameters, would be extremely useful. With ongoing efforts from experimentalists as well as theorists, the realization of SLIPS in everyday life should be possible in the near future.

**Author Contributions:** Conceptualization, V.K. and S.L.S.; writing—original draft preparation, D.T., P.R. and S.L.S.; writing—review and editing, D.T., P.R., A.V.S., V.K. and S.L.S. All authors have read and agreed to the published version of the manuscript.

**Funding:** V.K. and S.L.S. acknowledge the generous support received from BHU under the IOE scheme.

**Institutional Review Board Statement:** Not applicable.

**Informed Consent Statement:** Not applicable.

**Data Availability Statement:** Not applicable.

**Conflicts of Interest:** The authors declare no conflict of interest.

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
