# Peer review of "Durability of Slippery Liquid-Infused Surfaces: Challenges and Advances"

_coatings, doi:10.3390/coatings13061095_

Round 1
Reviewer 1 Report
This manuscript provides a review of the challenges and recent advances in the durability of SLIPS. The concept and work principle of liquid-infused surfaces were introduced concisely. The lubricant depletion and the durability of SLIPS were explained in detail. The structure of this manuscript s complete. Many newest advances in this field were cited. It provides a good reference for readers to study the concept, mechanism, and latest progress about the SLIPS. The reviewer believes this manuscript can be accepted after a minor revision as follows:
1. Some figures should be rearranged: the deformation of the font in Figures 2, 7, and 9 ; the font is too tiny in Figure 6 and 13;
2. The writing can be improved:
i) “A number of mechanisms can drive the depletion of the lubricant from the within the textures.” The “from the within the textures” cannot be understood.
ii) “The finest example of a liquid-infused slippery surface is found in Nepenthes Pitcher plant.” How to define an example as the “finest”?
3. There are too many mistakes in the use of “;”. It seems the “;” should be “:” in many following situations:
Line 20-21; line 87-89; line 107-109; line 128-129; line 158-159;
and so on.
4. The following papers provide a wonderful demonstration and application of the coexistence of Wenzel and Cassie state. They are recommended to be cited in the part of “1.1 wetting of a solid surface”
ACS applied materials & interfaces, 2019, 12(1): 1757-1764.
Advanced Materials, 2018, 30(31): 1802172.
Science Advances, 2021, 7(34): eabi7498.
The reviewer believes this manuscript can be accepted after a minor revision as follows:
1. Some figures should be rearranged: the deformation of the font in Figures 2, 7, and 9 ; the font is too tiny in Figure 6 and 13;
2. The writing can be improved:
i) “A number of mechanisms can drive the depletion of the lubricant from the within the textures.” The “from the within the textures” cannot be understood.
ii) “The finest example of a liquid-infused slippery surface is found in Nepenthes Pitcher plant.” How to define an example as the “finest”?
3. There are too many mistakes in the use of “;”. It seems the “;” should be “:” in many following situations:
Line 20-21; line 87-89; line 107-109; line 128-129; line 158-159;
and so on.
Author Response
"Please see the attachment."

Reviewer 2 Report
The overall review is comprehensive, but still major revision shall be made for the final publication:
Abstract:
The abstract provides a clear overview of the topic and the significance of SLIPS in various applications. But it would be helpful to mention the potential benefits or implications of enhancing lubricant retention abilities. And please using more objective expression and reduce the usage of I and we in the abstract.
Keywords:
Missing in current version, better provide 5 keywords accordingly
Introduction (Should be chapter 1 rather than 0):
1. Consider providing a brief explanation of wetting transitions and their relevance to SLIPS to improve the reader's understanding.
2. Elaborate on the mechanisms responsible for lubricant depletion, including any notable challenges or limitations.
3. Ensure the introduction flows smoothly by connecting the various points more cohesively.
4. Please cite the following papers for a better elaboration of the surface profiling as well as the slippery liquid phases
Coatings 2019, 9(9), 560; https://doi.org/10.3390/coatings9090560
Coatings 2020, 10(4), 377; https://doi.org/10.3390/coatings10040377
5. The wording legend for the Figures is a bit long, please reduce the word length
6. Some other wording and expression issue:
Instead of using "the liquid to be avoided" in Line 64, specify the type of liquid, such as "a low-surface tension organic liquid."
In Line 66, "micro-structured" can be replaced with "microstructured" for consistency.
Line 69 can be rephrased for clarity: "There is an increased risk of liquid sticking to the surface due to defects, irreparable damages, and contamination."
Instead of mentioning that there are several review articles in Line 71, provide a concise overview of the challenges, limitations, and advances made in the field of lotus leaf mimics.
In Line 73, "utilizes" can be replaced with "uses" for simplicity.
In Line 76, "overlying macroscopic wetting film" could be clarified as "a continuous film of liquid that covers the entire solid surface."
Line 79 can be rephrased: "The highly compressible air pockets are replaced with a liquid, which is comparatively less compressible."
Instead of "simple and complex liquids" in Line 83, consider using "a wide range of liquids" for clarity.
In Line 85, it would be better to list the applications separately and provide a brief description for each.
Line 88 can be improved for clarity: "The loss of lubricant due to evaporation, displacement, cloaking, or other mechanisms is a significant limitation."
In Line 90, "its" should be replaced with "the" to clarify the reference to the infused lubricant layer.
In Line 91, consider specifying the focus of the review article, such as "we will discuss recent studies on improving the durability of liquid infused surfaces."
The transition to the next section (Sec.2) can be improved to provide a clearer structure for the article.
Instead of using abbreviations like "Sec.2" and "Sec.3," consider writing "Section 2" and "Section 3" for better readability.
In Line 97, consider rephrasing for clarity: "We will also discuss recent case studies addressing the challenge of lubricant depletion."
In Line 101, "could be realize" should be "could be realized."
1. Liquid infused surfaces (Should be chapter 2 rather than 1):
Overall, the paragraph provides a good introduction to the topic of liquid-infused surfaces and their applications. However, there are a few areas where improvements can be made to enhance clarity and flow. Here are some suggestions for improvement:
In the first sentence, clarify what "Coatings 3 of 37" refers to. Is it a journal, a conference, or something else? Provide more context.
Instead of mentioning that certain topics are "out of the scope of this review" (line 58), consider removing those references altogether to maintain focus on the specific scope of the review.
In lines 61-63, rephrase the sentence to improve clarity. For example: "The roughness plays a critical role in minimizing the contact between the solid surface and external liquid by trapping air pockets within the surface protrusions. However, these air pockets are highly compressible and cannot withstand pressure, leading to a loss of water repellency under increased pressure or impact."
In line 68, change "robustness" to "robustness and durability" for clarity.
In lines 70-71, specify the context of the "excellent review articles" mentioned. Are they specifically related to lotus leaf mimics or liquid-infused surfaces in general? Provide more specific information.
In line 73, consider rephrasing "An alternate route" to "An alternative approach" for better readability.
In lines 80-81, clarify the meaning of "deposition happens on the lubricant layer." It may not be clear to the reader what is being deposited and why.
In line 85, rephrase "pose a promising candidate" to "are a promising option" to improve clarity.
In lines 88-89, rephrase "loss of lubricant due to evaporation, displacement, cloaking, or any other mechanism is the biggest limitations" to "The biggest limitation is the loss of lubricant due to evaporation, displacement, or other mechanisms." Additionally, consider mentioning "cloaking" earlier in the paragraph or provide a brief explanation of what it means.
In line 92, remove "In this review article" as it is already clear from the context.
In line 93, clarify what "the next section (Sec.2)" refers to. Is it part of the same review article or a different section/article?
In line 100, change "the the" to "the" for correct grammar.
In lines 105-106, rephrase the sentence for clarity. For example: "A liquid-infused surface consists of an underlying textured solid and a suitable liquid lubricant that can wick into, spread, and adhere stably within the protrusions."
In line 110, rephrase "the SLIPS maintain the repellency for liquids of varying surface tensions" to "the SLIPS maintain repellency even for liquids with varying surface tensions."
In line 119, consider removing "within a short span of time" as it doesn't add essential information.
In line 125, remove the extra "the" in "review the the fundamentals."
In line 138, rephrase "hydrophobic material will become more hydrophobic" to "hydrophobic materials will become even more hydrophobic."
In line 149, change "found by equating" to "obtained by equating."
In lines 150-151, clarify the meaning of "mixed state" and provide a brief explanation of what it entails.
2. Durability of SLIPS
It is recommended to provide a brief definition of what SLIPS are, since the term is used extensively throughout the paragraph without prior introduction.
Some of the sentences are quite lengthy and convoluted, which can make it challenging for readers to understand the key points being made. Therefore, it is suggested to break down some of the longer sentences into shorter, more concise sentences.
The paragraph could benefit from a clearer structure, with distinct sections and headings, to make it easier for readers to follow the main arguments being made.
It would be useful to provide specific examples of the practical applications of SLIPS and the industries where they are currently being used. This would help to provide context and illustrate the potential benefits of enhancing their durability.
The paragraph contains a lot of information, but it lacks coherence and clarity. It would be helpful to reorganize the content and provide clearer topic sentences to guide the reader.
Provide more context and background information at the beginning of the paragraph to introduce the topic of SLIPS technology and its purpose. Explain why resource wastage prevention is important and how SLIPS addresses this issue.
When introducing the figures, provide a brief explanation of what each figure depicts and its relevance to the topic. This will help the reader understand the content and purpose of the figures more easily.
Use clear and concise language throughout the paragraph. Some sentences are long and convoluted, making it difficult to grasp the intended meaning. Simplify the language and sentence structure where possible.
Provide transitions or logical connections between sentences and paragraphs to create a smooth flow of information. This will make it easier for the reader to follow the progression of ideas.
Check for grammatical errors and typos to ensure the paragraph is free from mistakes.
3. Summary and Future Prospects:
The paragraph is quite long and could benefit from being split into shorter, more focused sections.
The introduction of SLIPS is somewhat unclear, and it might be helpful to provide a brief explanation of what they are before launching into their potential applications and properties.
There are a few grammatical errors that could be corrected, such as the missing hyphen in "low surface tension liquid" and the misspelling of "retentivity".
The paragraph mentions several articles and studies, but it would be helpful to provide more information on what these studies found or how they contributed to the field.
The conclusion of the paragraph could be strengthened by highlighting some specific research questions that need to be addressed in order to develop SLIPS for commercial use, rather than just suggesting that more theoretical and computational studies are needed.
It might be useful to provide some concrete examples of how SLIPS could be used in various applications, rather than just mentioning that they have potential in many areas.
4. Conclusion
A short paragraph of conclusion shall be provided individually to summarize all the findings and reviews.
The overall quality of the english expression is not objective or academic enough. Please find a native english speaker or use professional language AI software for the polishing.
Author Response
"Please see the attachment."

Reviewer 3 Report
Comments and questions
1. Overall, almost all the figures are difficult to read. Some figures are too small and the text is not legible. In addition, the figures, which appear to be quotations, are not scaled down correctly and have been deformed so that they are taller than the original figures.
1-1) The figure is too small and needs to be corrected to a larger and clearer figure.
Fig.5, Fig.6, Fig.15, Fig.16
1-2) The figure is deformed vertically and needs to be corrected because it is not scaled down correctly from the original figure.
Fig.2, Fig.7, Fig.9, Fig.13
2. L1 on p. 1. says "Slippery liquid infused porous surfaces....." . Indeed, Fig.2 (c). gives the reader an image of a porous substrate with irregularly sized continuous pores.
On the other hand, p. 3, L105-107, shows "textured solid....... . within protrusions," and L127 on p.4 to L212 on p.8 describe general wetting phenomena when regular protrusions are placed on the surface of dense materials, which does not give an image of porous materials with continuous pores.
2-1) The analytical results of L127 on p. 4 to L212 on p. 8 are unlikely to be applicable to porous materials with irregularly sized connecting pores. The description of the wetting behavior of liquid against the surface of porous materials should be written more clearly.
2-2) The objective could be achieved with solids with regular protrusions on the surface instead of porous materials. If this is the case, the process of forming regular protrusions on the surface should be described more clearly, as well as the methods and problems involved.
Author Response
"Please see the attachment."

Round 2
Reviewer 2 Report
All the mentioned issues have been properly addressed. Thanks for the improvement.
N/A
Reviewer 3 Report
As review paper, this involves engineering significance. And the draft has been appropriately revised in response to my request, and is therefore approved for publication.